# Maternal health during the COVID-19 pandemic: Experiences of health workers in three Brazilian municipalities

**Ruth Helena de Souza Britto Ferreira de Carvalho**[1]*, **Maria Teresa Seabra Soares de Britto e Alves**[1], **Aluísio Gomes da Silva-Junior**[2], **Gisele Caldas Alexandre**[2], **Tatiana Raquel Selbmann Coimbra**[3], **Maurício Moraes**[4], **Letícia Oliveira de Menezes**[5], **Sandro Schreiber de Oliveira**[4,5], **Erika Barbara Abreu Fonseca Thomaz**[1], **Zeni Carvalho Lamy**[1], **Lely Stella Guzman Barreira**[3]

1 Public Health Department, Federal University of Maranhão, São Luís, MA, Brazil, 2 Fluminense Federal University, Niterói, RJ, Brazil, 3 PAHO/WHO, Brasília, DF, Brazil, 4 Federal University of Rio Grande, Pelotas, RS, Brazil, 5 Catholic University of Pelotas, Pelotas, RS, Brazil

* ruth.britto@ufma.br

**Data Availability Statement:** As this is a qualitative research, with in-depth interviews, addressing sensitive issues from the point of view of

## Abstract

### Objective

To analyze the experiences of maternal health workers in three Brazilian cities, located in the Northeast (São Luís), Southeast (Niterói), and South (Pelotas) regions during the first year of the COVID-19 pandemic.

### Methods

Qualitative research carried out between December 2020 and February 2021. Interviews were conducted, in person or remotely, with 30 health workers, doctors and nurses, working in maternity hospitals of different degrees of complexity.

### Results

Sociodemographic characteristics, employment relationships and professional qualification of the interviewees were described. Two thematic axes were identified: 1) changes in hospital organization and dynamics in the pandemic; 2) Illness and suffering of health workers. The majority of respondents were women. Most physicians had work relationships in the public and private sectors. In Niterói, health workers had better professional qualifications and more precarious work relationships (as temporary hires), compared to São Luís and Pelotas. In the context of the uncertainties resulting from the pandemic, this situation generated even more insecurity for those workers. The statements at the beginning of the pandemic covered topics such as changes in the organizational dynamics of services, healthcare, telemedicine, and interaction between health workers and users. In the health workers' perception, the initial period of the health emergency, which resulted in intense changes in the provision of services, was marked by an increase in preterm births, perinatal mortality, and fetal losses. Work overload, fear of contamination, concern for family members and uncertainties regarding the new disease caused intense suffering in health workers

identifying health workers, and, considering that the statements that support the findings and conclusions were made available in the paper, the authors believe that the availability of the entire transcription of the focus groups could violate the ethical precepts of guaranteeing the secrecy and privacy of the participants. So, all of the data (transcription of all speeches) have not been made publicly available, but several excerpts from the speeches of the health workers (de-identified) are inserted throughout the paper.

**Funding:** The authors are grateful for the technical and financial support of the Bill and Melinda Gates Foundation [INV-017424], World Health Organization (WHO) and Pan American Health Organization (PAHO) - ZCL, EBAFT, AGSJ, GCA, MTSSBA, RHSBFC, LOM, SSO, MM, YBM, TRSC, BBR, LSGB. Also, the National Council for Scientific and Technological Development (Conselho Nacional de Desenvolvimento Científico e Tecnológico – CNPq acronym in Portuguese) [processes 306592/2018-5 (EBAFT), 314939/2020-2 (ZCL), 311479/2020-2 (MRSSBA) and 308917/2021-9 (EBAFT)] and the Coordination for the Improvement of Higher Education Personnel (Coordenação de Aperfeiçoamento de Pessoal de Nível Superior – CAPES acronym in Portuguese) [finance code 001] (EBAFT, MTSSBA, RHSBFC, ZCL) for support for scientific publication.

**Competing interests:** The authors declare that they have no conflict of interest

who had little institutional support in the cities studied. The suffering experienced by health workers went beyond the work dimension, reaching their private life.

## Conclusion

Changes caused by the pandemic required immediate adjustments in professional practices, generating insecurities in healthcare regardless of the location studied. The method of hiring health workers remained the same as the previously practiced one in each city. Due to the risk of disease transmission, measures contrary to humanization practices, and more restrictive in São Luís, were reported as harmful to obstetric care. The Covid-19 pandemic was a huge challenge for the Brazilian health system, aggravating the working conditions experienced by health workers. In addition to the work environment, it was possible to briefly glimpse its effects on private life.

## Introduction

The COVID-19 pandemic quickly spread across most countries, causing deleterious effects on health services and, consequently, on health professionals working in these services. Even countries with well-structured health systems suffered the consequences of the work overload among health professionals and the challenges of reorganizing services and allocating medical resources during the pandemic period [1, 2].

During the COVID-19 pandemic, health workers became increasingly infected by the SARS-CoV-2 virus. The literature records a higher risk of death and greater lethality of health workers in areas with the highest occurrence of cases [3], as well as the consequences of the pandemic on the mental health of health workers [4, 5], resulting in changes in medical work [6].

The World Health Organization (WHO) / Pan American Health Organization (PAHO) had the "COVID-19: Mitigating indirect effects on essential health services for neonates, children, adolescents, and elderly people" initiative, with funding from the Bill Gates Foundation, supporting 20 priority countries in the world. In the Region of the Americas, Bolivia and Brazil participated. The initiative evaluated and encouraged the maintenance of essential services as a result of the COVID-19 pandemic. To this end, it was oriented to identify and contextualize actions to provide essential health services for vulnerable populations, and one of the priority areas was the care of women during pregnancy, childbirth and puerperium, and children [7].

This unprecedented global crisis has created substantial challenges for the healthcare of women during pregnancy, childbirth, and the postpartum period, particularly in the first year of the pandemic. And a significant concern has been the continuation of maternal and newborn health services, including antenatal, intrapartum, and postnatal care. Professionals involved in healthcare experienced changes in work processes and social restrictions with profound effects on their professional and personal relationships [8]. The results of these measures on health professionals were harmful and included high rates of infection and death, stress related to ignorance, fear, and uncertainty about the impact of the epidemic on the continuity of healthcare for non-COVID patients, as well as impairments in personal responsibilities, including caring for their families and themselves [9].

As to health workers, we sought to present the context in which health services were called upon to produce changes to adapt their health care offer to a specific population: pregnant and postpartum women. In this period, marked by so many uncertainties, the changes in the

obstetrics service organizational dynamics and their implications in the experiences of health workers are the object of this study.

The Brazilian National Health System (Sistema Único de Saúde—SUS), although structured at the national level, has important regional inequalities that affect its response capacity [10, 11]. At the beginning of the pandemic, the government response was uncoordinated and poorly structured on a scientific basis. The best government responses have been at subnational levels [11].

Disputes between narratives about the modes of transmission of Sars-Cov-2 that emerged at the beginning of the pandemic shaped political responses and brought dangerous consequences for infection prevention and control in terms of public health. Confrontations between scientific and moral narratives, the predominance of droplet transmission discourse, to the detriment of air transmission, in the dispute to support evidence-based decisions produced artifacts (posters, disinfectant dispensers, distancing markers) and social practices, including rituals of purification. In this context, the political ideology that favors individual choice and freedom sustained distrust and resistance to the isolation rules recommended by scientists and imposed by governments [12].

According to ABRASCO [11], the epidemiology of COVID-19 in Brazil was similar to what was happening in the world, but it also displayed important particularities. Between February 2020 and April 2022, 30 million cases and 660,371 deaths attributed to COVID-19 were confirmed, generating an accumulated incidence rate of 14,163.6 cases per 100,000 inhabitants, and an accumulated mortality rate of 311.7 deaths per 100,000 inhabitants. There was a decrease in the provision of prenatal consultations throughout Brazil in 2020, which caused enormous damage [13]. Brazil also had the highest rates of maternal mortality from COVID-19 worldwide. In 2021, more than half of the deaths of pregnant women or postpartum women (59%) diagnosed with COVID-19 were not associated with factors of previous risk or comorbidities, suggesting a relation with the inefficiency of the health systems and the country's inability to manage the pandemic. Some factors associated with these deaths were: delays in testing and identifying disease-related symptoms, delays in hospitalization after diagnosis of COVID-19 and, finally, lack of intensive care after hospitalization [14].

In this perspective, a multicenter study was carried out with health workers from maternity hospitals in three Brazilian municipalities, as part of the "COVID-19: Mitigating indirect effects on essential health services for neonates, children, adolescents, and elderly people" initiative, in Brazil. Counties considered as having well-coordinated responses during the COVID-19 pandemic, with different sociodemographic characteristics and great diversity both in geographical and cultural aspects, and in the structure of health services and intersectoral network, and located in the Northeast (São Luis), Southeast (Niterói), and South (Pelotas) regions, were chosen. This approach allowed comparing the effects of the pandemic on the work of doctors and nurses, identifying similarities and differences in women's healthcare services in the first year of the health emergency.

This study aims to analyze the experiences of maternal health workers in São Luís, Niterói, and Pelotas during the first year of the COVID-19 pandemic. Exploring the work and existential experiences of maternal health workers in the context of a health emergency contributes to understanding the challenges faced in healthcare and offers subsidies to Health Policies aimed at mitigating problems and advancing the quality of maternal and child care.

## Methods

Qualitative research with in-depth interviews based on comprehensive theory [15], carried out from December 2020 to February 2021, in maternity hospitals in three different Brazilian

regions, Northeast, Southeast, and South, located respectively in the municipalities of São Luís (MA), Niterói (RJ), and Pelotas (RS). The diversity in the characteristics of the maternity hospitals aimed at expanding the universe of the study and analyzing different settings.

In São Luís, a public maternity hospital of high complexity was included, a benchmark in the healthcare of high-risk pregnant women and newborns with intensive care needs, in the state. The institution offers comprehensive healthcare to women and children, from specialized prenatal care to childbirth and postpartum follow-up. It has 16 high-risk pregnancy beds, 20 for gynecological hospitalization and 52 for rooming-in. At the beginning of the health emergency, it was a benchmark in the healthcare of COVID-19 cases. In economic terms, the municipality presents the following indicators: the average monthly salary of formal health workers is 3.1 minimum wages, and the human development index (HDI) corresponds to 0.768. The municipality has a population of 1,115.932 inhabitants/km$^2$ [16].

In Niterói, a public maternity hospital for normal risk pregnancy, a benchmark for childbirth in the municipality, but not a benchmark for COVID-19 healthcare, was selected. The institution has 22 surgical obstetric beds, an Intermediate Neonatal Unit, and does not offer outpatient services. The municipality ranks first in terms of per capita income in the state of Rio de Janeiro, with a high- and middle-income population and an HDI of 0.886. The population size is 513,584 inhabitants/km$^2$ [16].

In Pelotas, there were four hospitals for pregnancy. But only two maternity hospitals were included in this study because only those two hospitals were open during the pandemic to care for patients through SUS. The first one is a benchmark health facility in the municipality and macro-region in various specialties and was a benchmark for patients with COVID-19. The second one is a benchmark service in maternal and child care, and besides the maternity hospital, the institution has pediatric and neonatal intensive care units. In 2020, the average monthly salary was 2.8 minimum wages and the HDI was 0.739. The municipality is medium-sized, with a population of 343,826 inhabitants/km$^2$, and a regional benchmark in the state [16].

## Study participants and sampling

Doctors and nurses who worked in the maternity hospitals were selected based on a nominal list of all professionals who worked in the obstetric hospitalization sectors during the initial period of the pandemic, provided by directors and heads of services. A purposeful sampling was defined, seeking to include workers with a diversity of these characteristics. A total of 30 health workers were interviewed, 10 in São Luís, 12 in Niterói, and 08 in Pelotas. Considering the large number of professionals working in the investigated institutions, the final number of respondents was established by data saturation. Therefore, the interviews were interrupted when the responses began to repeat information already obtained [17]. The saturation point was determined in workshops, in each municipality, during the period of the interviews, to identify the moment of repetition of the themes. Interviews were carried out after repetitions were observed, seeking confirmation of saturation.

## Data collection

After identifying the workers to be interviewed, contact was made by telephone and/or face-to-face for presenting the research and inviting participation. There were no refusals in Niterói. In São Luís and Pelotas, in the case of indirect refusals, the participants were replaced by workers with similar characteristics, based on the nominal list provided by the head of the sector.

Two tools were used. A questionnaire structured with sociodemographic and professional trajectory data, such as qualifications, time of experience, and employment relationship, with open answers and multiple choice. The second tool was an interview, recorded and later

transcribed by fellows in the research group. A semi-structured research script on issues related to experiences during the COVID-19 pandemic was used. Although the data were collected after the first wave of the pandemic in the three cities, the script questions addressed the initial moment of the health emergency. This document included changes in the environment and in the work routine, in the supply and demand for services, in the availability and adequacy of equipment, and in the perception of risk by health workers, as well as the meanings attributed to the disease, and the security measures adopted at work and in private life, as shown in the attached script.

The team of researchers, composed of public health workers, two in São Luís, two in Pelotas, and three in Niterói, all with experience in qualitative data collection, conducted the interviews in person, by telephone, or digital platform, on the days and at the times agreed in advance with the health workers. Workshops were held to develop and discuss the single script and align the interviewers to seek accuracy and consistency between interviewers [18], as well as clarity and pertinence of script questions. The analyzes of each city were presented and discussed in a joint workshop with all researchers. The interviews, with an average duration of 50 minutes, were recorded and transcribed in full.

## Analysis

The content analysis was carried out in the thematic modality proposed by Bardin [19] and Minayo [20]. The steps for carrying out the thematic analysis were: pre-analysis and exploration of the collected material; data processing and interpretation. In this way, after exhaustive reading of the interviews, the data were coded, to extract the relevant themes and then interpret the content, linking the lines with the context of their production [20]. The results are presented based on themes related to changes in the work environment and in the relationships between health workers and users, and to the experience of suffering and illness of health workers. The article was written based on COREQ recommendations [21].

The situations mentioned by the interviewees concern a period of exceptionality identified as the most critical, due to the speed of changes caused in the work environment and in private life. Our theoretical-methodological proposal consisted of understanding the meanings attributed by subjects in social interaction [15], placing the context as a total social fact [22], i.e., as a phenomenon that connects several domains such as social and individual aspects, as well as physical and psychic aspects.

## Ethical considerations

The research was approved by the Research Ethics Committee of the University Hospital of the Federal University of Maranhão, under CAAE number 35645120.9.0000.5086, in compliance with resolution 466/12 of the National Health Council and by the Research Ethics Committee of the Pan American Health Organization, PAHOERC Ref. No. 0260.02. All participants signed a free and informed consent form (TCLE, acronym in Portuguese). To guarantee the confidentiality and anonymity of the health workers interviewed, their names were suppressed and replaced by the city of origin, gender, professional category, and the number of occurrences of the interview.

## Results

### Healthcare professionals: Sociodemographic characteristics, employment relationships and professional qualifications

The 30 health workers interviewed were distributed into the following categories: 17 doctors and 13 nurses. The majority were women, about 83.4%. In the total sample, the age ranged

from 30 to 64 years old. In the three municipalities, there was a predominance of participants self-identified as white, followed by brown. Niterói health workers had a shorter time working in the service (median 1.5 years) and more precarious work relationships. Temporary workers did not have their labor rights guaranteed.

In São Luís, the service uptime in the institution was longer and, as in Pelotas, they had more stable employment relationships, supported by labor legislation. Most health workers, especially doctors, had other work relationships in the public and/or private sector. In terms of professional training, São Luís had the highest number of health workers with higher education, with two of them with a master's degree. The South and Southeast regions had a total of six health workers with a master's degree and three with a doctorate degree. The number of academic titles did not guarantee a more stable job for the professional in the sample. Workers linked to a municipal healthcare institution had lower job guarantees than those working in university hospitals.

### Themes of the statements

The results are presented in two thematic axes: Changes in hospital organization and dynamics in the pandemic, and Illness and suffering of health workers.

**Changes in hospital organization and dynamics in the pandemic.** In this section, we will address how the measures to face the pandemic produced changes in the organization of space and in the provision of services in the investigated health facilities. From the perspective of professionals, the interruption of elective care, the closing of beds, the creation of new areas aimed at assisting patients with Covid-19 and the establishment of new care rules marked a new institutional dynamic. The threat posed by the health emergency also led to changes in the use of work tools, such as personal protective equipment (PPE) and collective protection equipment (CPE), and in the relationships between professionals and users. Concern was also reported about the consequences of limiting access to patients without suspicion or diagnosis of COVID-19. The relationship between risks and rights was also the object of reflection by the health workers, as well as their assessments of the past moment.

> "*Everything changed! The physical areas have changed. Our service protocols, equipment, and clothing have changed.*" (São Luís, male, doctor 8)

At the beginning of the pandemic, the hospital structures available in the maternity hospitals studied were deemed insufficient and inadequate by health workers for the healthcare and isolation of pregnant and parturient women with suspected or diagnosed COVID-19. In Pelotas and São Luís, specialties and elective procedures were suspended. Beds were closed to make entire wards available for the isolation of pregnant and postpartum women.

> "*Obstetrics was, in fact, the only service in the hospital that remained open, attending patients on free demand and with the emergency room opened*". (São Luís, male doctor, *5)*

> "*All surgical beds were closed, and the residents were relocated to other rooms, which were eventually interdicted as well. We had 12 beds closed so that the COVID maternity unit could be opened.*" (Pelotas, female nurse 4)

> "We started the first phase [of covid-19], which was the phase when we had the most patients, with 11 beds, right? So, this had a huge impact, because there were 11 wards that closed and each one could only hold one patient." (São Luís, male doctor 1)

In the three maternity hospitals, healthcare rooms were opened for patients identified as having flu-like symptoms. Spaces such as the lounge for health workers and the recreational

space for the team have become inappropriate due to the new protocol recommendations for contagion mitigation. In Niterói, new wards were opened in a site that was deactivated for a previously planned renovation. A professional meeting room was adapted for another use, such as a patient waiting room. The reduction of spaces for socialization of health workers, increasing the isolation among them, was also noted. In terms of impact on the operation of services, the low-risk maternity had fewer changes compared to the others.

> "*It is a low-risk maternity. (. . .) We haven't changed much, no, and people keep coming. We did not have to close the outpatient clinic, there were no services that we had to stop* [. . .] *relocation of people from a larger area that was not COVID-19 to a COVID-19 area.*" (Niterói, female doctor 1)

Due to the emergence of that moment, changes in the organizational dynamics of services and service spaces occurred quickly and with the imposition of new service rules. The reorganization of the hospital space imposed by the sanitary emergency produced new practices related to contamination control and protection of health workers and patients.

> "*At the beginning, we were still discussing: do we only put a mask* [on] *those who have symptoms or does everyone have to wear a mask? So, after we had all things very clear concerning PPE, everything was very calm.*" (São Luís, male doctor 5)

Initially, the changes were accompanied by a lack of knowledge and uncertainties regarding the need for use and the type of equipment to be adopted in each situation. Once the initial period of lack of consensus among experts on procedures to avoid contamination had passed, the use of greater amounts of equipment began to be adopted in health institutions.

*Changes in protocols*. Changes in work processes, according to the statements, indicated the creation and application of protocols, with systematic training on the healthcare of patients suspected of COVID-19. The main trainings were on donning and doffing and routine environment cleaning after caring for a patient. The interval between appointments for pregnant women increased because a disinfection was mandatory before and after each procedure.

> "*We had to make a change in the routine [of the service] . . . give more time for patients. Instead of attending in a three-hour interval (. . .) we had to put a six-hour interval, to give them time to arrive, have an interval between one and the other, clean the place, all that. (. . .) The patients cannot sit next to each other at the reception (. . .). So a lot has changed!*" (São Luís, female doctor 6)

However, according to health workers, the pregnant women and their companions complained about this delay and perceived it as neglect and lack of attention, and not as a careful measure to prevent the spread of the disease. Initially, the creation of an isolation space for patients with suspected virus infection was not accompanied by the increase in the number of health workers in the sector, making service difficult, as an adaptation to PPE use was required to care for the patients. In São Luís, this procedure was considered time-consuming and a delaying factor.

> "*Including the training, you had to go there and put on the gown, put on a mask, all very neat, the order in which the equipment is put on and taken off.*" (São Luís, male doctor 8)

In general, there was availability and adequacy of personal (PPE) and collective (CPE) protective equipment, instructions for use, training, and implementation of protocols, aiming at the protection of health workers and service users.

"*Here at the maternity ward, it was very good, there were no shortages of PPE, there was no shortage of respirators, it was very good in relation to these outfits.*" (Niterói, female doctor 1)

"*I joined the service and I have already been called for training on how to wear PPE: donning and doffing, why the hygiene or the mask*". (Pelotas, female doctor 2)

However, in Pelotas, there were differences in the adoption of training and in the acquisition of personal protective equipment. Initially, in one of the maternity hospitals, not considered a benchmark, there was insufficient PPE and training, forcing health workers to purchase their own safety equipment.

The protection measures, whose purpose was to function as barriers to the transmission of the virus, also caused adverse effects, hampering the daily healthcare routine by producing, in addition to physical distancing, difficulties in communication between health workers and patients. In São Luís, the health workers considered the availability of adequate PPE, but reported that the use of masks and face shields, in particular, made communication between health workers and users difficult. The equipment was considered uncomfortable and heavy, and its continuous use left sores on their faces, but above all, it rendered listening difficult.

"*There was a significant communication difficulty, both when speaking or listening, because this* [face shield] *causes an echo and communication is bad, regardless of whether you are talking to a patient or a teammate. It's uncomfortable, it hurts. I think that, from the point of view of the doctor-patient relationship, there was, indeed, a certain distancing from each other.*" (São Luís, male doctor 5)

In Pelotas, they reported discomfort from the use of the equipment and the occurrence of caustic lesions and allergies on their faces, on some occasions. PPE use training was reported to be efficient.

*Between risk and good health practices.* In general, the threat of the risk of infection affected the relationships between health workers and patients, reducing their interaction. The teams were concerned with the reception of patients and the quality of the services offered, but also with their own safety and that of the patients.

*Now, in the patient who is a suspect, everyone is always a little afraid to get close to her. So, we realize that they end up, unintentionally, but they feel. . . a little. . . isolated. Because there's no contact, we don't get near them all the time. So, they feel a little of this distancing between the professional and the patient*". (Niterói, female, nurse 4)

According to interviewees from the maternity hospitals studied, a distancing of health workers and patients suspected of infection, in particular, was noticed by the latter, who experienced greater isolation. In Niterói, difficulty in adherence to preventive measures by patients and their families was also mentioned.

The safety recommendations against COVID-19, at the beginning of the pandemic, contradicted the recommendations for the humanization of childbirth. Given the risk, some rights guaranteed by law, such as the presence of a companion of the woman's free choice, including during childbirth, were suspended in São Luís. In Pelotas and Niterói, the permission of one single companion was adopted for pregnant women. Walking restrictions and mandatory use of a mask by the patient at the time of delivery were also among the measures taken by health workers in São Luís who realized how much the new practices contradicted the previous ones,

causing suffering to parturients, their families and health workers who attended them. This situation was described as a contradiction in terms of good practices.

> "*So, creating isolation to protect mothers who were not symptomatic, isolating this symptomatic woman, isolating and explaining this isolation to this companion was very complicated. . .And we had to explain to her that she had to wear a mask all the time. We didn't have a bathroom just for her. It was horrible, because we were in this duality, in this contradiction (. . .). The patient in labor, for example, one of the measures we took was to ask her to walk around. How was she going to walk*?" (São Luís, female nurse 1)

At the beginning of the pandemic, tests for COVID-19 were not available to all women in São Luís, but all who were hospitalized were monitored as a suspected case of COVID-19. At the time, there were few tests available, and the result took days to be released. The Niterói maternity hospital offered testing for COVID-19 (rapid test and PCR).

> "*It's a room only for those who were suspected cases, with main signs, an isolated service. The ward was also only* for those who had signs and symptoms. We started doing tests, *the rapid test, and the RT-PCR test, according to the onset time and the signs and symptoms according to the complaint*". (Niterói, female doctor,2)

In Pelotas, the availability of tests for COVID-19 was also insufficient in the two maternity hospitals studied, especially in the one that was not a benchmark for pregnant women suspected or diagnosed with COVID-19.

*Health care*: Changes and effects. A strategy adopted to relieve the reduced number of health workers on leave due to illness or comorbidities and the partial suspension of services was the implementation of telemedicine. The health workers in São Luís considered this practice inadequate and of low effectiveness, because, usually, the prenatal care of pregnant women requires face-to-face contact and physical examination.

> "*We were not sure about managing the patient only by telemedicine, without them coming to be evaluated here.*" (São Luís, male doctor 1)

In addition, they feared that communication difficulties in the service at-a-distance model would be even greater than in face-to-face consultations. In Pelotas, this strategy was used to maintain the continuity of outpatient care and monitoring of pregnant women, especially those with COVID-19.

In two cities, São Luís and Pelotas, there was a perception of an increase in the demand for prenatal care in maternity hospitals. According to interviewees, this fact was related to the decrease in the offer of consultations in the public primary healthcare network. As a result, there was an increase in the number of pregnant women with few prenatal consultations in maternity hospitals. The difficulty in maintaining continuity of healthcare for women was related to changes in the healthcare provided by primary healthcare units.

> "*Goodness*! *it's nine months generating a being without any exam, without any consultation, without any laboratory, without any ultrasound image. Nothing nothing nothing.. So, no vaccine.*"

> [. . .] "[the pregnant woman] *didn't come because she didn't want to come. She was afraid. And she didn't come because she even wanted to, but there was no service for her anywhere. Everything was COVID, COVID, COVID*". (São Luís, female nurse 9)

In the first months of the health emergency, in São Luís, the public primary care facilities were divided into COVID-19 units and non-COVID-19 units, with the relocation of users from one healthcare facility to another. This change did not generate good results, because the information was not well publicized and also because, according to the health workers interviewed, women avoided leaving their homes. In Pelotas, care at public primary care facilities took place in separate shifts, but even so, there was a low capacity of solving problems, because one single healthcare shift was not enough to meet the needs of users.

In the emergency service of the high-complexity hospital in São Luís, in addition to this increase in demand, the occurrence of complaints commonly resolved in routine prenatal consultations was also identified. In Pelotas, an increase in the flow of pregnant women was reported in maternity hospitals that were not a benchmark for COVID-19, and this increase was explained by the women's fear of getting contaminated in these maternity hospitals. The same happened in Niterói.

[. . .] "*what actually changed was the demand. We had a demand equivalent to 60–70 deliveries per month, and this number of deliveries changed a lot, we reached 120–130 deliveries per month.*" (Niterói, female nurse 4)

The maternity hospital studied absorbed the impact of the demand from other maternity hospitals. In all municipalities, difficulties were reported for women in accessing routine vaccination and necessary tests.

Hospitalization in the COVID-19 ward was described as one of many delicate situations. In São Luís, pregnant women resisted being isolated, claiming that the place caused them anxiety, and their companions were reluctant to leave them alone. To minimize the suffering caused to women hospitalized by the restrictions imposed, there was a greater offer of psychological care. Professionals from São Luís reported that the initial period of the pandemic was marked by an increase in preterm births, perinatal mortality and fetal losses, and difficulties in scheduling a puerperal consultation.

"*We also had more complications. I don't know how many. . ., but occasionally a patient with PPD* [probable date of delivery] *would come, more severe cases were coming because the patients stayed at home.*" (São Luís, female doctor 4)

Retrospectively, health workers from all the maternity hospitals studied considered the adjustments to be positive for meeting the needs of coping with the pandemic, even though the reduction of beds and interruption of healthcare in other clinical and surgical sectors were mentioned. The suspension of activities that took place in the most critical period of the pandemic, from March to July 2020, created a pent-up demand for services in other specialties in Pelotas and São Luís. In Pelotas, the health workers interviewed expressed concern about the pent-up demand for services, such as gynecological surgical procedures and outpatient care. In São Luís, consultations and surgeries were resumed in August, on a reduced basis to ensure a presumedly safe resumption of services. At the time of the interview, in December, a manager who worked in healthcare stated: "*There are still a lot of people in the line*". According to him, patients were also returning at a slow pace because of a decline in economic conditions during the pandemic, an impoverishment caused mainly by the loss of jobs.

"*We were not able to clear all the lines, for us to get back to normal. There are still a lot of people in line. And coming to the hospital, in the capital, implies a cost, right?*" (São Luís, male doctor 1)

This research was carried out at a time when there were still no vaccines, but there was already greater knowledge accumulated about the disease and somehow health workers believed that the pandemic was starting to recede. Elective services, pent up due to the temporary suspension, were being resumed, and the pace of transmission was falling, as well as the number of deaths. These health workers spoke as if the most difficult phase of the pandemic was already in the past.

**Illness and suffering of health workers.** In this section, we will address how the statements of the health workers about their experience in the first year of the pandemic were marked by the intense psychological suffering caused by the changes that have occurred, both in the context of hospital care and in the context of social life. Professional insecurity to deal with an unknown disease, the risk of illness and absence from work, contamination of family members, fear, and loss of loved ones, in addition to patients, all these elements and even more the social transformations resulted in feelings of sadness, pain, indignation and isolation.

At the beginning of the pandemic, changes in the work process and safety protocols generated anguish and insecurity even in health workers accustomed to working in maternity hospitals. The requirement of physical distancing, due to the elevated risk of contamination, and the lack of knowledge about the new disease caused ambiguous situations that mixed fear, conflict and cooperation.

*Work overload, lack of workers and their effects.* Many interviewees, especially physicians, worked in both the public and private sectors. At the beginning of the pandemic, the interruption of activities in private healthcare clinics and medical offices resulted in greater availability of health workers for assistance in the public sector, in São Luís. In a brief time, however, this situation was reversed to a lack of health workers. In Niterói, the accumulation of jobs in public services caused work overload and a greater risk of exposure for some of the health workers, doctors and nurses interviewed.

To meet the need for skilled labor, one of the administrative management strategies consisted of relocating health workers from various specialties to care for COVID-19 patients and, in the case of maternity hospitals, from gynecology to obstetrics. The lack of expertise to work on a little-known disease contributed to a feeling of insecurity shared at that time by health workers from São Luís and Pelotas.

The removal of health workers with comorbidities and, above all, illness due to COVID-19 resulted in work overload in the remaining health workers (30 to 35% in São Luís and about 25% in Niterói). After the relocation, another institutional action was a proposition to increase the workload of health workers. This situation resulted in changes in service schedules, further intensifying the workload of these health workers.

"*There was no hiring at first. There was only an extension of the workload for those who wanted it.*" (São Luís, female doctor 3)

"*Leaves were suspended, vacations were suspended, because of COVID-19. . . there was an administrative direction in this sense, of having people on standby to cover absent people.*" (Niterói, female nurse 1)

In the four maternity hospitals studied, some of the changes were the suspension of vacations and work permits, i.e., health workers' rights, to maintain the workforce in that emergency. In the three municipalities, the changes were intense in the maternity hospitals and the work overload was often mentioned as another reason for illness.

"*People were working much harder than they could.*" (São Luís, male doctor 1)

The lack of health workers, in general, was highlighted, as well as the delay in hiring and the new hires quitting soon after taking up their positions. In general, the public administration carried out temporary contracts and tenders on an emergency basis to replace health workers on leave due to illness and comorbidities. However, in Pelotas and São Luís, this effort did not have a satisfactory effect, as the institutional rules required for selection and admission of new hires did not supply the immediate need for replacements. Another problem is that some of the new hires did not have the necessary experience for an intense job like the one experienced in healthcare at that time, to the point of quitting the job shortly after their hiring. In Niterói, hiring took place as needed, but as temporary work, without labor guarantees.

*Ambiguities*: *Fear*, *conflict*, *cooperation*. At the beginning of the health emergency, illness of health workers and changes in the intensity of work were described as a moment of great stress, but also of a lot of cooperation and solidarity among health workers who, when reporting this moment, were often surprised by the positive result of the work carried out even with teams that were reduced in size for months.

> "*And the number of medical leave certificates, the number of sick leaves increased, resulting in a huge work overload in the team, as it is quite difficult for us to have a shift with the full team, because one has a medical certificate, the other is away for some reason, some lost family members. . . and we have been experiencing this kind of stress daily.*" (Niterói, female nurse 3)

> "*There were shifts in which I had several technical staff employees absent* [. . .] *it was a miracle*" (São Luís, female nurse 9)

Fear of contamination was felt among health workers at that time. According to the statements, older health workers or those with comorbidities, due to the greater risk of complications, in case of contamination, suffered even more from the dilemma of fulfilling their professional duties in the face of the risk of exposure to the virus. This context greatly affected the relationship of health workers with patients. Resistance, insecurity and even the refusal of health workers to care for women diagnosed with COVID-19 aggravated situations of conflict between managers and healthcare workers. Such ambiguity mixed fear, insecurity, and ethical duty, producing effects that hampered healthcare.

> "*When there was an older colleague on the team who stated, "I'm not going to do that because I'm too afraid of contaminating myself. So, there was this dilemma, a real Sophie's choice: I have to do it, but I'm afraid to do it. It was an experience that brought much anguish to many people, many colleagues (. . .) in our way of caring, of seeing the patient, of seeing the other. I think that, in a way, this eliminated empathy.*" (São Luís, male doctor 5)

Another element of tension concerns the use of protocols that included drugs without scientific evidence for the treatment of patients with a confirmed diagnosis of COVID-19. This fact was reported especially in São Luís and was the cause of conflicts among doctors and between them and the hospital management, and these clashes are yet another aggravating factor in the relationship among health workers.

> "*I was following the Ministry of Health protocol, which did not provide for the use of hydroxychloroquine and other medications. . . . and that, sometimes, generated a certain conflict because someone wanted to do it. . . And also, as there was nothing closed, it caused me doubts and insecurity.*" (São Luís, male doctor 1)

The institutional support for the treatment of diseases acquired in the work environment, and other psychological problems resulting from that situation, was deemed insufficient, requiring them to seek psychological support individually in the private sector.

> "*I was the first to get COVID and I didn't get any help from the hospital. I didn't have access to exams, I didn't have access to a CT scan, I had to pay it in a private service. I mean, I contaminated myself in the work environment and had no assistance, you know?*" (São Luís, female doctor 2)

In Niterói, the considerable proportion of temporary employment relationships among health workers was pointed out as an element that contributed to even greater insecurity in terms of financial and emotional stress. The solidarity of teammates was reported as a great support for coping with the situation experienced.

The rate of illness among health workers due to COVID-19 varied among the cities studied, but the reported suffering did not vary in intensity, frequency, or origin. Fear of contagion and illness, insecurity associated with the disease, ignorance, stress, tension, resistance, cooperation, and solidarity were elements that intertwined and marked the experiences described.

*Risk perception and safety strategies: Between work and home.* The hospital environment represented risk and concern of health workers with their families. Various strategies have been created to prevent the spread of the virus in the home environment. Although some institutions have proposed paying for daily rates in hotels close to the maternity hospitals, our informants, mostly women, chose to redouble their care and stay at home. In the home environment, other spaces and care protocols were created to enable the professional to live with his family members, even at a distance. All these efforts did not alleviate the feeling of responsibility and guilt for putting their own families at risk. The distancing of older relatives, such as parents, was another point of suffering.

> "*On May, on Mother's Day, I took a 15-day leave to be able to see my parents and they stayed there for another 15 days fulfilling a strict isolation. It was very difficult to stay away from my children to preserve them.*" (Pelotas, female doctor 2)

Hostilities, open or veiled, suffered by health workers in public or private spaces, such as residential condos, also contributed to the feeling of isolation. The statements of the health workers from the maternity hospitals studied reveal the distance between media appreciation and the daily stigmatization that occurs in face-to-face interactions.

> "*I was the person who. . . the COVID nurse!! And then* [. . .] *really* [. . .] *I was excluded from everything* [. . .] *I was excluded from family gatherings* [. . .] *not that they were always getting together, you know?*" (Pelotas, female nurse 3)

The speeches of health professionals consist of overlapping feelings such as sadness, anxiety, uncertainty, fear and revolt caused by the combination of elements that range from experiences of work overload, physical exhaustion, and difficult medical decisions in a context of little evidence from scientific studies, to risks and fears regarding the disease transmission, including experiences of private life marked by isolation and guilt for the possibility of transmission of the disease. In the case of Pelotas and São Luís, health workers reported an increase in alcohol and psychotropic medication consumption in that context.

## Discussion

Our study on the perspective of professionals, about their experiences, revealed that in large maternity hospitals, the interruption of elective care and the closing of beds resulted in a damming of demand for services. There was also an increase in the number of complications in childbirth, due to the decrease in prenatal care in public primary care units and the fear of pregnant women leaving home and attending public health units. In all the institutions investigated, new areas were created aimed at assisting users with Covid-19. To control contamination, new service protocols were implemented, such as the use of protective equipment by professionals and users, there was also a restriction on the rights of users to companions, changing good practices. To compensate for the decrease in the number of professionals, due to illness and comorbidities, telemedicine was adopted in São Luís and Pelotas. In Niterói, the usual-risk maternity and non-reference for Covid-19 care had increased demand for deliveries.

In the reports about the work routine, the uncertainties caused by the lack of knowledge about the disease, the ethical conflicts between the fear of falling ill, of transmitting the disease to the family and professional duty were sources of suffering for professionals in São Luís, who also felt helpless by the institution when they got sick. Internal conflicts between professionals and users and family members also caused suffering. However, if fear increased conflicts, professional commitment produced cooperation and critical reflections on the measures adopted. The use of face shields, for example, produced barriers to virus transmission and also to communication between professionals and users. Telemedicine did not offer safety to professionals, since prenatal care requires a physical examination, but it was adopted as a follow-up for pregnant women with Covid-19.

In addition to the work scope, the suffering extended to private life. Perceptions of risk also influenced strategies to protect family members, showing the effects of work during the pandemic on private life.

### Social dramas: The pandemic as a total social fact

Experiences, feelings, thoughts and social actions are linked to each other. The COVID-19 pandemic can be framed in what Marcel Mauss [22] called a total social fact, a phenomenon that crosses several domains of society (economic, scientific, political, legal, religious, and family), linking individual and social aspects, on one side, to physical and psychological aspects, on the other [12].

The health emergency, due to the speed of the changes provoked at a global level, constituted a period of exceptionality in social life. Regarding the perception of events, Leach [23] stated that some social changes represent a continuity, a passage, which marks time as a succession of facts, whereas others are experienced as disruptive elements, bringing transformations that break with the established patterns. The pandemic was experienced as a rupture on several levels: in the public sphere, in the market, in work, in knowledge, in practices, in interpersonal relationships; in the private sphere, family relationships and intimacy [24].

The pandemic also contradicted the experience of illness in the 20th century. Before modern life, the fear, anguish, and death associated with illness were linked to a contagious disease such as epidemics. The evil was collective, the disease came from contact with the other. In that context, the only way to prevent and avoid contagion was the isolation of the sick [25].

In the modern world, the reality and imagery of the disease have lost their collective character. The disease became individualized. In industrial societies, it is the individual who is sick and suffering is linked to chronic and degenerative diseases. Even though there were epidemic and endemic diseases, they no longer represented a danger, a threat to humanity.

The Covid-19 pandemic had a paradoxical effect: we were experiencing a new and unexpected situation for the modern world and at the same time we had concepts, such as isolation and distance, to combat an epidemic [26]. Why we were not capable of adopting isolation and information to fight this evil that devastated us and imposed so much material damage and so many subjective transformations is a question that will take some time to answer, as well as understand its multiple effects on our lives.

To understand the emotions experienced and the multiple meanings attributed to the health emergency, it is necessary to place this phenomenon, of global scope, in the context of its occurrence [26]. Emotions (fear, guilt, anguish, indignation, among others) can be understood as a language through which the tensions caused by ignorance, risk, threat, and uncertainty were communicated and experienced at that moment [27–29].

## Narratives in dispute

The aforementioned narrative dispute about the modes of transmission of Sars-Cov-2 shaped political responses and brought dangerous consequences for infection prevention and control in terms of public health. In this context, the political ideology sustained distrust and resistance to the isolation rules recommended by scientists and imposed by governments [12].

The pandemic forced us to be distanced and isolated, as the threat of the disease was in objects and people. Rituals to purify the body and surfaces reinforced the dominant narrative by establishing boundaries between "clean" and "contaminated" [12].

Health workers, despite the use of individual and collective protective equipment, were contaminated, and many became ill and also died as a result of the disease. The work of these health professionals was considered potentially dangerous not only because of self-contamination, but also for others. Our interviewees suffered because they considered themselves a threat to their families, but also because they were stigmatized.

Contrary to the manifestations in the media in which they were praised as heroes, health workers reported, either in an anecdotal tone or with resentment, the avoidances and hostilities suffered by being identified as potential contaminants of the Sars-Cov-2 virus.

In Brazil, the Unified Health System (SUS) has been suffering for some years a process of lack of funding that compromises both human resources and the provision of inputs and, consequently, the quality of the services offered. During the pandemic, this situation has become even more aggravated. Even tougher working conditions were imposed on health professionals, such as work overload and physical and mental illness [30]. Once the pandemic is under control, timid efforts have been made to modify the conformation of the SUS.

## COVID-19 and the damage to maternal and child health

Globally, the COVID-19 pandemic caused changes in health services and affected the quality of care for women and newborns at childbirth and during the puerperium, with a reduction in the supply of health services for women and newborns, and a consequent fall in indicators: such as the increase in the number of stillbirths, the medicalization of birth, the increase in the number of cesarean surgeries, the induction of labor, maternal anxiety and stress and decreased family participation [31].

In the Brazilian scenario, the pandemic represented a huge challenge for the Unified Health System (SUS) and the response varied among states and municipalities, and the most vulnerable in socioeconomic terms were the most affected ones [10]. There were similarities in the way the municipalities studied responded to the pandemic as to the adoption of measures such as the lockdown. There were also difficulties in accessing primary healthcare and maternity hospitals, testing, examinations, and vaccines, hampering maternal and child healthcare.

Changes in the flow of patients in primary healthcare services, the delay in identifying the group of pregnant women as at risk, as well as recommendations for physical isolation, made it difficult for pregnant women to access prenatal care in the first months of the pandemic, and they, in many cases, flocked to maternity hospitals, overloading healthcare. Chisini et al. [13] found a decrease in the provision of prenatal consultations throughout Brazil in 2020 and the enormous damage caused.

In the case of hospitalization, women were followed up as suspected cases of COVID-19, but not all of them were tested, due to difficulties in accessing testing, as was also evidenced in other Brazilian states, at the beginning of the pandemic [32]. Due to the restriction of circulation, ensuring at least one companion for the pregnant or postpartum woman was seen as a guarantee of the humanization guidelines and served to reduce the women's stress.

In the three cities studied, maternal health care was described based on measures related to the reorganization of healthcare flows, adaptation of environments, training of health workers, availability of supplies, as well as the relationships among health workers, users, and family members, in maternity hospitals located in three cities in Brazil. Such strategies, fundamental for mitigating the effects of the pandemic, were narrated by health workers willing to share their experiences in a period marked by many uncertainties.

## Impact of the pandemic on medical work

Sacristán & Millán [33] warned about four aspects that deserve attention to better understand the impact of the pandemic on medical work. The first one concerns the frequent publication of fake and sensationalist news. The second concerns the risks of making medical decisions not based on evidence, at a time when scientific evidence was still scarce or contradictory. The third is related to the bioethical implications when the therapeutic resources available are not enough for everyone. And finally, the fourth is the need to rethink medical learning and the use of technologies such as telemedicine to overcome certain difficulties in care.

In our research, we identified two of the four aspects addressed by the authors. Medical decisions are often based on scientific evidence. At times, such as the beginning of the pandemic, when such evidence was scarce, contradictions emerged in the face of the lack of evidence on the effectiveness of drugs, as well as on the risk of their adoption. The narrative disputes around this theme went beyond the technical dimension, causing divergences and aggravating the conflict among health workers.

The overlap of information, scientific evidence and fake news greatly contributed to a sense of helplessness experienced especially by those who were overloaded with work in sectors considered essential, such as hospitals.

The feminization of the health workforce, especially the fact that the largest contingent of health professionals and health workers in the sector is composed of women, draws attention to the accumulation of working hours [34], especially during the pandemic.

Conflicts occurred in the team regarding compliance with the treatment protocol of people hospitalized with a confirmed diagnosis of COVID-19, due to the use of medicines without scientific evidence. Santos-Pinto et al. [35] discussed the seriousness of this problem in Brazil, insofar as the use of medicines without evidence in the treatment and prevention of COVID-19 was encouraged [36].

Health workers commonly face a substantial risk of exposure to infectious diseases, the pandemic, however, greatly increased the number of health workers affected by the disease, as pointed out by Teixeira et al. [34]. Numerous cases of deaths of physicians by COVID-19 were reported in the world, mainly between April and May 2020. The countries that most report deceased physicians in Latin America were Brazil (n = 113) and Ecuador (n = 110) [3]. By the

beginning of 2021, Brazil had suffered more than 200,000 deaths from COVID-19, among whom 500 were nursing health workers, who worked on the front lines of the battle against the virus. Europe had the highest absolute numbers of reported infections (119 628) and deaths (712), but the Eastern Mediterranean region had the highest number of reported deaths per 100 infections (5.7) [37, 38].

Research conducted during the first wave of the pandemic, in Italy, highlighted the enormous psychological and physical impact on health workers, with reports of frequent psychic and somatic symptoms. Measured emotional exhaustion levels were significantly higher than usual values found in other Italian samples before the COVID-19 outbreak [39]. The high prevalence of mental illness among health workers during the pandemic was also corroborated by Pappa et al. [40], in a systematic review, and by Nasi et al. [41], in qualitative research with nursing health workers. Psychological support for health workers was discussed by Meleiro et al. [42]. Other study evidenced that Brazilian healthcare professionals showed aspects of quality of life that were more affected during the COVID-19 pandemic like alcohol and psychotropic medication consumption [43].

The pandemic has exacerbated the need for a policy for the development of human resources in health that values planning, regulation of labor relations and the permanent education of health professionals and health workers in the sector, contrary to what has been observed in the daily management of the SUS at the federal, state, and municipal levels [34].

## Strengths and limitations of the study

The study included a few maternity hospitals, but some were benchmark maternity hospitals for COVID-19 patients and did a great job in serving this population. Comparisons among us should be analyzed with caution. In São Luís, we only included workers from a high-risk benchmark facility. In the other municipalities, medium-complexity hospitals were also included. Despite limiting comparisons, this strategy allowed us to explore particularities of services among hospitals in different Brazilian regions. Some interviews were carried out face-to-face and others were mediated by technology (online), which may have, in some way, limited interactions between researcher/interviewee and generated some communication noise due to possible connection difficulties; but, on the other hand, they allowed greater security for both, given the seriousness of the pandemic situation at the time, and did not impair the quality of the interview.

This is the first qualitative study, using triangulation of methods and data collection techniques, in a multicentric approach, interviewing health professionals (doctors and nurses) working at hospitals with different levels of obstetric risk, caring for women with and without a history of infection by COVID -19 during pregnancy or hospitalization for childbirth. The study allowed to investigate aspects of the work and private life of health workers based on the reporting of their experiences during the 1st year of the pandemic. The research addressed professional practices carried out in three Brazilian cities with different geographical, economic, and social settings.

It is known that qualitative research does not seek, a priori, generalization. Therefore, care must be taken when interpreting and applying the results in different contexts and populations. The transferability of results depends on factors such as geographic diversity, adequate selection of representative participants, validity and reliability of data collection and analysis methods, and clarity of description of the research context. Thus, in this study, when carrying out the research in high and low-complexity maternity hospitals, in three cities in different regions of Brazil, including women with and without covid, with different social realities, the transferability of the results can be considered by including different contexts and participants

who brought the diversity of experiences and perspectives, making the results potentially more representative.

## Conclusion

The pandemic affected the continuity of care for pregnancy, delivery, and birth. In crisis situations, changes at various levels in health management occurred. It is essential to understand the challenges experienced to face problems in maternal and child healthcare services.

The organization of isolated physical spaces for the healthcare of sick and symptomatic patients, differentiated circulation flows and the implementation of work safety routines were important measures in the organization of services. However, these measures had the impact of overloading the work of health workers and losses in the humanization of childbirth processes.

Safety protocols against COVID-19 need to be made compatible with the guidelines for the Humanization of childbirth to prevent the mental suffering of pregnant women and loosen up the relationship between health workers and women, as observed in the municipalities studied.

Undersized teams and administrative difficulties to replace those on leave and sick, in a timely manner, generated tensions, intensification of work and mental suffering of health workers. The management of the workforce demands reflection on the part of managers in their management, especially in times of crises such as the pandemic. It also needs more agile legal provisions to make it possible to hire and replace personnel.

The provision of care through telemedicine did not obtain the expected result, being criticized by health workers. Its use should be rethought, alternating face-to-face and virtual moments, as well as preserving bonds between health workers and their patients. Continuing Education processes can reduce difficulties in the use of communication technologies.

The intense mental suffering reported deserves attention and a differentiated approach to planning work processes and supporting health workers and management aiming at mitigating and preventing this problem.

## Acknowledgments

The authors would like to thank the health professional who participated in this research.

## Author Contributions

**Conceptualization:** Ruth Helena de Souza Britto Ferreira de Carvalho, Maria Teresa Seabra Soares de Britto e Alves, Erika Barbara Abreu Fonseca Thomaz, Zeni Carvalho Lamy, Lely Stella Guzman Barreira.

**Data curation:** Ruth Helena de Souza Britto Ferreira de Carvalho, Maria Teresa Seabra Soares de Britto e Alves, Aluísio Gomes da Silva-Junior, Gisele Caldas Alexandre, Sandro Schreiber de Oliveira.

**Formal analysis:** Ruth Helena de Souza Britto Ferreira de Carvalho, Maria Teresa Seabra Soares de Britto e Alves, Aluísio Gomes da Silva-Junior, Gisele Caldas Alexandre, Maurício Moraes, Letícia Oliveira de Menezes, Sandro Schreiber de Oliveira, Zeni Carvalho Lamy.

**Funding acquisition:** Tatiana Raquel Selbmann Coimbra, Lely Stella Guzman Barreira.

**Investigation:** Ruth Helena de Souza Britto Ferreira de Carvalho, Maria Teresa Seabra Soares de Britto e Alves, Aluísio Gomes da Silva-Junior, Gisele Caldas Alexandre, Maurício

Moraes, Letícia Oliveira de Menezes, Sandro Schreiber de Oliveira, Erika Barbara Abreu Fonseca Thomaz, Zeni Carvalho Lamy.

**Methodology:** Ruth Helena de Souza Britto Ferreira de Carvalho, Maria Teresa Seabra Soares de Britto e Alves, Aluísio Gomes da Silva-Junior, Gisele Caldas Alexandre, Tatiana Raquel Selbmann Coimbra, Maurício Moraes, Letícia Oliveira de Menezes, Sandro Schreiber de Oliveira, Erika Barbara Abreu Fonseca Thomaz, Zeni Carvalho Lamy, Lely Stella Guzman Barreira.

**Project administration:** Tatiana Raquel Selbmann Coimbra, Lely Stella Guzman Barreira.

**Resources:** Tatiana Raquel Selbmann Coimbra, Lely Stella Guzman Barreira.

**Supervision:** Aluísio Gomes da Silva-Junior, Tatiana Raquel Selbmann Coimbra, Maurício Moraes, Sandro Schreiber de Oliveira, Lely Stella Guzman Barreira.

**Validation:** Ruth Helena de Souza Britto Ferreira de Carvalho, Maria Teresa Seabra Soares de Britto e Alves, Aluísio Gomes da Silva-Junior, Gisele Caldas Alexandre, Tatiana Raquel Selbmann Coimbra, Maurício Moraes, Letícia Oliveira de Menezes, Sandro Schreiber de Oliveira, Erika Barbara Abreu Fonseca Thomaz, Zeni Carvalho Lamy, Lely Stella Guzman Barreira.

**Visualization:** Ruth Helena de Souza Britto Ferreira de Carvalho, Maria Teresa Seabra Soares de Britto e Alves, Aluísio Gomes da Silva-Junior, Gisele Caldas Alexandre, Maurício Moraes, Letícia Oliveira de Menezes, Sandro Schreiber de Oliveira, Erika Barbara Abreu Fonseca Thomaz, Zeni Carvalho Lamy, Lely Stella Guzman Barreira.

**Writing – original draft:** Ruth Helena de Souza Britto Ferreira de Carvalho, Maria Teresa Seabra Soares de Britto e Alves, Aluísio Gomes da Silva-Junior, Gisele Caldas Alexandre, Maurício Moraes, Sandro Schreiber de Oliveira, Erika Barbara Abreu Fonseca Thomaz, Zeni Carvalho Lamy.

**Writing – review & editing:** Ruth Helena de Souza Britto Ferreira de Carvalho, Maria Teresa Seabra Soares de Britto e Alves, Aluísio Gomes da Silva-Junior, Gisele Caldas Alexandre, Tatiana Raquel Selbmann Coimbra, Maurício Moraes, Letícia Oliveira de Menezes, Sandro Schreiber de Oliveira, Erika Barbara Abreu Fonseca Thomaz, Zeni Carvalho Lamy, Lely Stella Guzman Barreira.

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
