## [Decision Letter · Decision Letter 0]

23 Jan 2023

PONE-D-22-34357Maternal health during the COVID-19 pandemic: experiences of health workers in three Brazilian municipalitiesPLOS ONE

Dear Dr. Carvalho,

Thank you for submitting your manuscript to PLOS ONE. After careful consideration, we feel that it has merit but does not fully meet PLOS ONE’s publication criteria as it currently stands. Therefore, we invite you to submit a revised version of the manuscript that addresses the points raised during the review process.

We look forward to receiving your revised manuscript.

Kind regards,

Kyaw Lwin Show, MPH

Academic Editor

PLOS ONE

Journal Requirements:

"The authors are grateful for the technical and financial support of the Bill and Melinda Gates Foundation [INV-017424], World Health Organization (WHO) and Pan American Health Organization (PAHO) - ZCL, EBAFT, AGSJ, GCA, MTSSBA, RHSBFC, LOM, SSO, MM, YBM, TRSC, BBR, LSGB. Also, the National Council for Scientific and Technological Development (Conselho Nacional de Desenvolvimento Científico e Tecnológico – CNPq acronym in Portuguese) [processes 306592/2018-5 (EBAFT), 314939/2020-2 (ZCL), 311479/2020-2 (MRSSBA) and 308917/2021-9 (EBAFT)] and the Coordination for the Improvement of Higher Education Personnel (Coordenação de Aperfeiçoamento de Pessoal de Nível Superior – CAPES acronym in Portuguese) [finance code 001]  (EBAFT, MTSSBA, RHSBFC, ZCL) for support for scientific publication."

"The authors would like to thank the financial support of the Pan American Health Organization (PAHO) [CON20-00012173]. Also, the National Council for Scientific and Technological Development (CNPq, acronym in Portuguese) [processes 306592/2018-5, 314939/2020-2, and 308917/2021-9] and the Coordination for the Improvement of Higher Education Personnel(CAPES, acronym in Portuguese) [finance code 001] for support for scientific publication.."

"The authors are grateful for the technical and financial support of the Bill and Melinda Gates Foundation [INV-017424], World Health Organization (WHO) and Pan American Health Organization (PAHO) - ZCL, EBAFT, AGSJ, GCA, MTSSBA, RHSBFC, LOM, SSO, MM, YBM, TRSC, BBR, LSGB. Also, the National Council for Scientific and Technological Development (Conselho Nacional de Desenvolvimento Científico e Tecnológico – CNPq acronym in Portuguese) [processes 306592/2018-5 (EBAFT), 314939/2020-2 (ZCL), 311479/2020-2 (MRSSBA) and 308917/2021-9 (EBAFT)] and the Coordination for the Improvement of Higher Education Personnel (Coordenação de Aperfeiçoamento de Pessoal de Nível Superior – CAPES acronym in Portuguese) [finance code 001]  (EBAFT, MTSSBA, RHSBFC, ZCL) for support for scientific publication."

7. We note that you have stated that you will provide repository information for your data at acceptance. Should your manuscript be accepted for publication, we will hold it until you provide the relevant accession numbers or DOIs necessary to access your data. If you wish to make changes to your Data Availability statement, please describe these changes in your cover letter and we will update your Data Availability statement to reflect the information you provide.

**Additional Editor Comments:**

It is crucial to adhere to a reporting checklist, such as COREQ, in order to ensure explicit and comprehensive reporting in qualitative studies.

Reviewers' comments:

Reviewer's Responses to Questions

**Comments to the Author**

1. Is the manuscript technically sound, and do the data support the conclusions?

Reviewer #1: Yes

Reviewer #2: Partly

Reviewer #3: Partly

Reviewer #4: Yes

2. Has the statistical analysis been performed appropriately and rigorously? 

Reviewer #1: N/A

Reviewer #2: N/A

Reviewer #3: N/A

Reviewer #4: N/A

3. Have the authors made all data underlying the findings in their manuscript fully available?

Reviewer #1: No

Reviewer #2: No

Reviewer #3: No

Reviewer #4: No

4. Is the manuscript presented in an intelligible fashion and written in standard English?

Reviewer #1: Yes

Reviewer #2: No

Reviewer #3: Yes

Reviewer #4: Yes

5. Review Comments to the Author

Reviewer #2: Thank you for the opportunity to review this paper. The objective was to analyze the experiences of maternal health workers in three Brazilian cities, located in the Northeast, Southeast and South regions. Qualitative survey research carried out over a 3-month period - between December 2020 and February 2021. Interviews were conducted, in person or remotely, with 30 health workers, doctors and nurses, working in maternity hospitals of different degrees of complexity.

The study is interesting and it is good to see experiences from South America reported. I had a number of issues with the paper though as detailed below.

The objective in the abstract does not mention anything about the COVID-19 pandemic. The findings have the pandemic as a focus but this needs to be in the objective.

In the Abstract background the regions are initially given – Northeast, Southeast and South regions but then these terms are used - Niterói, São Luís and Pelotas. Are they the same?

The abstract specifically mentions hiring practices - The method of hiring health workers remained the same as the previously practiced one in each city – the reason is not clear. Is this related to the experiences?

The conclusion in the abstract introduces new issues including the private life impacts. Conclusions should summarise and provide implications for others rather than new information.

The opening sentence needs more context and dates - The COVID-19 pandemic quickly spread across several countries. I would also argue that the impact was not just in ‘several’ countries – surely it was in all countries?

The aim or objective of the study needs to be included at the end of the Introduction and needs to match with the objective in the Abstract.

I am a bit confused with the methods selected- Qualitative, exploratory survey research. This was predominantly an interview study which some survey data on demographics collected. Why choose to call it survey research?

The interviews addressed the ‘initial moment of the health emergency’ but the interviews were carried out from Dec 2020 to Feb 2021. Given so much happened in 2020, I am concerned that the initial moment of the health emergency could not really be collected?

A team of researchers, composed of public health professors and graduate students, conducted the interviews. Given there were only 30 interviews, how many interviewers were they? What was the process for consistency between many interviewers?

How were the doctors and nurses who worked in the maternity hospitals selected?

The analysis section is too brief. I do not understand what this sentence means either – ‘Content analysis was performed in the thematic modality’. I am also not familiar with thematic axes – do you mean themes?

The Discussion section is interesting but I struggled to find a link to the actual findings of the study. I would have liked to see a summary of the findings at the beginning (rather than a rewording of the focus of the study) and then a discussion of how the findings were similar or different to other research. At the moment, the Discussion seems to be quite separate from the study findings.

Reviewer #3: Thank you for the opportunity to review this article, I deeply appreciate the time and effort the research team has placed in the conduction of this study, writing of this manuscript, and preparing it for submission. This study provides insight on the experiences of maternity care during the early stages of the covid19 Pandemic in Brazil, it can potentially inform future interventions and programs for the improvement of maternity care services during infectious disease outbreaks. The manuscript as a whole should follow standard guidelines for reporting qualitative research as stated in PLOS ONE publication criteria website( https://journals.plos.org/plosone/s/criteria-for-publication) : “ Qualitative research studies should be reported in accordance to the Consolidated criteria for reporting qualitative research (COREQ) checklist or Standards for reporting qualitative research (SRQR) checklist. Further reporting guidelines can be found in the Equator Network's Guidelines for reporting qualitative research.” Please re-structure and expand the manuscript to follow the above guidelines. Below are specific recommendations for the manuscript text.

Introduction

While this section provides some background of international healthcare services issues related to covid19, it should be expanded to provide evidence on the context that is specific to Brazil regarding health service disruption, its impact on health workers, and maternal health outcomes during the covid19 pandemic. The evidence provided in the discussion section “ Covid-19 and the damage to maternal and child health” such as Tasca et al 2022, Chisini et al 2021, and Silva et al, 2021 would be better placed in the introduction so readers outside of Brazil can understand the significance of the study at the start of the manuscript.

Methods

It seems a survey was only used for the demographic characteristics of participants, but the majority of the data was obtained from in-depth interviews, if so, please clarify this by changing the first sentence of this section from “qualitative exploratory survey” to “qualitative in-depth interviews.”

It is necessary to use and explain a conceptual or theoretical framework guiding the methods and interpretations of findings (e.g. phenomenology, grounded theory, ethnography, etc.)

It is necessary to provide more information on the research team and reflexivity, such as but not limited to: which author/s conducted the interviews? What experience or training in research have they received? Did they have any relationship to the study participants prior to the study? Is it possible that the occupation, gender, or any other position of power the interviewer had influenced the answers participants provided or the way the data was collected and analyzed?

To improve clarity and the flow of ideas, in the last paragraph the sentence in interview duration (lines 160-161) would be best placed on the “Data collection section” and the informed consent sentence (lines 161-162) would be best placed in the “Ethical consideration section:”

Study techniques and tools

For clarity and avoiding redundancy I suggest removing this sub-heading and move the first paragraph of this section regarding study tools to the “Data Collection” section, and the second paragraph regarding ethics committee approval and confidentiality to the “Ethical consideration” section.

Data collection

This section needs more detail to assure the reader the research was conducted with scientific rigor. :

Please specify the dates data collection started and ended

Was the structured questionnaire open answers or multiple choice?

What questions or prompts were used during the interviews?

Was the interview pilot tested?

Where interviews recorded and transcribed? Who did the transcription?

Where the interviews done in more than one language or just Portuguese? Please state all languages

Where the transcripts returned to participants for any additional comments?

Were interview notes taken? Were they used for the analysis as well?

Data analysis

Please describe how the authors performed the thematic analysis, what steps were taken and by whom? Did more than one person code? If so, how were disagreements resolved? Was any software used or was it done manually?

Was there any author reflexivity during the analysis? E.g. How were the researchers assumptions and biases addressed/considered in the analysis?

Study participants and sampling

The phrase “intentionally defined sample” is not common use in Academic English, please correct to” purposeful sampling” if that is what the authors meant.

The same for “ saturation of the senses”, please correct this to “ data saturation”

Please explain why data saturation was used to determine sample size, and how it was determined that saturation had been reach.

How many participants dropped out before being interviewed? How were they replaced? Where participants interviewed more than once? Was there a follow up interview?

Results

This section needs to be extensively revised to follow best practice when reporting results of a qualitative analysis in academic English. The authors might find the following similar studies useful to re-write this section:

-Leung, C., Olufunlayo, T., Olateju, Z. et al. Perceptions and experiences of maternity care workers during COVID-19 pandemic in Lagos State, Nigeria; a qualitative study. BMC Health Serv Res 22, 606 (2022). https://doi.org/10.1186/s12913-022-08009-y

- Hazfiarini A, Akter S, Homer CSE, Zahroh RI, Bohren MA. 'We are going into battle without appropriate armour': A qualitative study of Indonesian midwives' experiences in providing maternity care during the COVID-19 pandemic. Women Birth. 2022 Sep;35(5):466-474. doi: 10.1016/j.wombi.2021.10.003. Epub 2021 Oct 11. PMID: 34656517; PMCID: PMC9239738.

Healthcare professionals: sociodemographic characteristics, employment relationship and professional qualifications:

-It is best practice to provide this information in the form of a table and to only summarize in text important information (such as the women being majority) or interesting patterns found in this data.

-The religion of the health workers does not seem to be relevant to the research objectives, please remove unless the authors can provide justification of its inclusion in the context of this research study.

Thematic axes of statements

-Please state at what point of the research process where the findings translated to English

Table 1 and table 2: Headers are needed for each column on these tables. It is unclear if each row represents a code, or a higher-level sub-category or sub-theme, please clarify. Additionally, there is repetition of both the content of column one and the column containing quotes, please revise and summarize to avoid redundancy.

To improve readability, please add subheadings that represent the subcategories or subthemes under each one of the two major themes 1) changes in hospital organization and dynamics of the pandemic and 2) Illness and suffering of health workers.

All statements in the findings need to derive directly from the data collected, therefore, each statement needs to be supported by relevant in-text quotes, context for the findings should be provided in the introduction or methods, and the valuable insights by the authors should be saved for the discussion and backed up by evidence which needs to be referenced. While all this section is very interesting, it is not clear what is commentary by the authors and what was reported by the participants.

Line 298 to line 317. Since this is a qualitative study looking at the experiences of health workers, be mindful that the findings cannot state factual increases in service demand. This can be demonstrated by hospital records and other quantitative data, which is beyond the scope of this research. This study can report in the results that participants perceived an increase in demand, and the authors can choose to compare this perception with evidence from other research looking into hospital records and statistics from that period and elaborate on the differences or similarities in the discussion section.

Discussion:

- I suggest the authors make use of this guide provided by the journal to revise this section to be more in line to common practice https://plos.org/resource/how-to-write-conclusions/

- The section “social dramas: the pandemic as a total social fact” provides valuable theoretical background which should be summarized and placed in the methods section.

- The sections “narratives in dispute” and “covid-19 and the damage to maternal and child health” and “the impact of the pandemic in the medical work” provide relevant context and background to the study, they should be summarized and incorporated to the introduction.

- Please revise this section to discuss in depth how the findings of this study specifically compare or relate to the available evidence and social theories provided in the mansucript

Reviewer #4: Thank you for the great opportunity in reviewing this manuscript. This is a very important study with valuable findings. Below are recommendations to improve the clarity and transparency of the research:

Abstract

• I wonder what precarious work relationships health workers om Niterói experienced compared to São Luís and Pelotas. Would be good to add one or two examples in brackets for this particular sentence (i.e., xxx, xxx).

Introduction

• In the second paragraph, one would wonder what “in this context” refers to, perhaps paraphrase this to be more clearer? For example, if authors refer to the COVID-19 pandemic, then authors can say “During the pandemic”

• It would be great if authors can give background information about COVID-19 in Brazil (i.e., burden, trends) to provide context for the readers, and then justify why looking at this in Brazil is important

• Adding information on the context of maternal health in Brazil (i.e., use of antenatal care, maternal health indicators – morbidity and mortality, the ratio of health workers to the population) both before and during the pandemic is critical for the readers to be able to interpret the results. I suggest adding this information too

• Providing information on how care changes throughout the pandemic from other settings will be helpful as well to illustrate for readers the kind of information this study was looking for

• These two studies have a good example of introduction and will be really great to look at when revising the introduction:

o Hazfiarini A, Zahroh RI, Akter S, Homer CSE, Bohren MA. Indonesian midwives' perspectives on changes in the provision of maternity care during the COVID-19 pandemic: A qualitative study. Midwifery. 2022 May;108:103291. doi: 10.1016/j.midw.2022.103291. Epub 2022 Feb 26. PMID: 35279435; PMCID: PMC8881222.

o Hazfiarini A, Akter S, Homer CSE, Zahroh RI, Bohren MA. 'We are going into battle without appropriate armour': A qualitative study of Indonesian midwives' experiences in providing maternity care during the COVID-19 pandemic. Women Birth. 2022 Sep;35(5):466-474. doi: 10.1016/j.wombi.2021.10.003. Epub 2021 Oct 11. PMID: 34656517; PMCID: PMC9239738.

Methods

• Please justify why interview was used compared to other methods (i.e, focus group discussion)

• Please justify the rationale behind choosing these three municipalities compared to others – what is this sampling based on? Is it because they have the highest COVID-19 rates? Or other reasons?

• Line 122 what does SUS stand for? Please use full abbreviations for the first use

• Are the two hospitals from Pelotas public hospitals? Please state this. And if it is, does that mean all public hospitals? If yes, it’d be good to provide justification for why only public hospitals are included

• Other than doctors and nurses, are there any criteria imposed for participation? It’d be great to be clear on the eligibility criteria

• Line 146, please add an explanation of how data saturation was determined

• Please add more details on the recruitment

o How did the recruitment happen? Did all doctors and nurses in the three hospitals were approached? Were advertisement materials circulated?

o How did the authors ensure the sociodemographic of participants are varied? Did this happen before approaching the participant or before?

o Did all the doctors and nurses in the three hospitals take the demographic survey first and then based on the survey results 30 doctors and nurses approached and recruited?

o When did the survey happen? Is this right before the interview? Or days before the interview itself?

• Please add more details on the data collection

o Please add details on the type of questions/themes asked during the interview and if interview guide/study instruments were piloted. Please attach interview guide/study instruments as an appendix

o How many people were involved in the interview? Is it only one person interviewing or there is one interviewer and one observer when the interview happens?

o Add information about where the interview was conducted (if it is facility-based or not) and discuss how the different settings might influence the participant's response

o In what language the interview happened? And please add how translations were done if it is in a non-English language

• Please add more details on the data analysis

o How many researchers were involved in data analysis?

o Was the codebook developed? Please add coding tree to the manuscript as an appendix

o Were the themes developed inductively or deductively?

o Was any framework used?

o Was any data management software (i.e., Nvivo, Atlas) used?

• Please add a section on reflexivity, such as information on how interviewer characteristics could influence the participant’s response

• Please add details on how many people refuse or drop out of the study

• Please ensure that the reporting of the study adheres to reporting standards for qualitative inquiry, I recommend the use of COREQ reporting standards for qualitative research to ensure compliance with reporting the methods and results of the study: Allison Tong, Peter Sainsbury, Jonathan Craig, Consolidated criteria for reporting qualitative research (COREQ): a 32-item checklist for interviews and focus groups, International Journal for Quality in Health Care, Volume 19, Issue 6, December 2007, Pages 349–357, https://doi.org/10.1093/intqhc/mzm042

Results

• Line 195, please add more details about what precarious work relationships mean and example of this

• Line 197, what is CLT? Please spell out abbreviations on first use

• I feel that the themes under each thematic axes can be categorized further, this will be really helpful when presenting too so readers will not be lost in words as further categorizations mean that sub-headings under thematic axes can be used, and this can guide readers better. For example, on the thematic axes 1: Changes in hospital organization and dynamics in the pandemic, the themes can be further categorized into services changes, human resource changes, protocol changes, etc. And the results section can be presented using the sub-headings of these categorizations

• Please use sub-headings when talking about different themes under the thematic axes to avoid confusion. Another alternative is to bold the specific themes when first use in the specific paragraph, for example: “In Pelotas and São Luís, specialties and elective procedures were suspended. Beds were closed to make entire wards available for the isolation of pregnant and postpartum women. In the three maternity hospitals, healthcare rooms were opened for patients identified as having flu-like symptoms.” I think further categorization of these themes into specific domains will be really helpful for readers as suggested on the previous points

• 232-233 are too short to stand by themselves as one paragraph, please merge them with another paragraph, same with 219 to 221.

• Please use quotations from participants to illustrate and provide grounds for interpretation. Please see the two studies mentioned in the introduction above for example

Discussion

• Before going to the social dramas heading, please add one paragraph summarising the findings of the studies, for example (taken from other study): “Our findings show that midwives in Surabaya and Mataram, Indonesia, strived to deliver maternity care during the COVID-19 pandemic. Despite the difference in the numbers of people with COVID-19 between the study sites, midwives in Surabaya and Mataram shared similar experiences in providing care, except for an increase in workload, which was mostly faced by midwives in Surabaya.” (Hazfiarini, 2021)

• The notion of social fact by Marcel Mauss is very interesting and it will enrich the discussion if authors can expand/connect the social fact define in the discussion with the study findings, and what lesson can be taken to prepare better for future pandemics

• Again, for narratives in dispute, what lessons can be taken from the study results? What can we do better for the health workers?

• Please add the strengths and limitations of the studies as “Strengths and limitations” section, and I think the strengths aspects can be expanded more Please add how the authors have addressed the identified limitations.

• Another aspect of the study is to discuss the “transferability” of the results to other settings in Brazil or beyond

6. PLOS authors have the option to publish the peer review history of their article (what does this mean?). If published, this will include your full peer review and any attached files.

Reviewer #1: **Yes: **Jen Sothornwit

Reviewer #2: No

Reviewer #3: No

Reviewer #4: No

---

## [Author Response · Author response to Decision Letter 0]

25 Apr 2023

Dear Academic Editor and Reviewers,

Thank you for the careful evaluation of the above-mentioned manuscript. After careful reading, we considered all suggestions made and responded to all the comments/suggestions in this document. We are resubmitting the manuscript to Plos One, addressing the points raised during the review process. In addition to the ‘Rebuttal letter’, we have also included a ‘Revised Manuscript with Changes Highlighted’ and an unmarked version of the paper without tracked changes, according to the instructions received.

Thank you very much for the opportunity to review our work.

 Kind regards,

 The Authors

Reviewer #2: 

The objective in the abstract does not mention anything about the COVID-19 pandemic. The findings have the pandemic as a focus, but this needs to be in the objective. In the Abstract background the regions are initially given – Northeast, Southeast and South regions but then these terms are used - Niterói, São Luís and Pelotas. Are they the same?

We accepted the suggestion and modified the text of the objective to address the questions raised by the reviewer.

“to analyze the experiences of maternal health workers in three Brazilian cities, located in the Northeast (São Luis), Southeast (Niterói) and South regions (Pelotas) during the first year of COVID-19 pandemic.”

The abstract specifically mentions hiring practices – the method of hiring health workers remained the same as the previously practiced one in each city – the reason is not clear. Is this related to the experiences?

We modified the text of the article to address the questions raised by the reviewer.

“In Niterói, health workers had better professional qualifications and more precarious work relationships (as temporary hires), compared to São Luís and Pelotas. This situation generated even more insecurity in those workers.” 

The conclusion in the abstract introduces new issues including the private life impacts. Conclusions should summarize and provide implications for others rather than new information.

We accepted the suggestion and modified the text of the article to address the questions raised by the reviewer 

The opening sentence needs more context and dates – The COVID-19 pandemic quickly spread across several countries. I would also argue that the impact was not just in ‘several’ countries – surely it was in all countries?

We accepted the suggestion and modified the text of the article to address the questions raised by the reviewer 

“The COVID-19 pandemic quickly spread across most countries, causing deleterious effects on health services and, consequently, on health professionals working in these services.”

The aim or objective of the study needs to be included at the end of the Introduction and needs to match with the objective in the Abstract.

We accepted the suggestion and modified the text of the article to address the questions raised by the reviewer 

“This study aims to analyze the experiences of maternal health workers in São Luis, Niterói and Pelotas during the first year of COVID-19 pandemic.”

I am a bit confused with the methods selected – Qualitative, exploratory survey research. This was predominantly an interview study which some survey data on demographics collected. Why choose to call it survey research?

We apologize for the error that occurred when translating the article into English. We did not carry out a survey research, but a qualitative research that used in-depth interviews.

The interviews addressed the ‘initial moment of the health emergency’ but the interviews were carried out from Dec 2020 to Feb 2021. Given so much happened in 2020, I am concerned that the initial moment of the health emergency could not really be collected?

We agree that many changes occurred throughout 2020, but the interview script was prepared with questions that referred to the peak period of the pandemic and we sought to make it clear that the objective was to understand the experiences related to the changes that occurred to face the pandemic. As this period was very significant for health workers, it was possible to recover this information, although the collection started in December 2020.

We have modified the text of the article to make this point clearer:

“Although the data were collected after the first wave of the pandemic in the three cities, the script questions addressed the initial moment of the health emergency, including changes in the environment and in the work routine, in the supply and demand for services, in the availability and adequacy of equipment, and in the perception of risk by health workers, as well as the meanings attributed to the disease, and the security measures adopted at work and in private life.”

A team of researchers, composed of public health professors and graduate students, conducted the interviews. Given there were only 30 interviews, how many interviewers were they? What was the process for consistency between many interviewers?

We rewrote the paragraph and included the answers to these questions: 

“The team of researchers, composed of public health workers, two in São Luís, two in Pelotas, and three in Niterói, all with experience in qualitative data collection, conducted the interviews in person, by telephone, or digital platform, on the days and at the times agreed in advance with the health workers. Workshops were held to develop and discuss the single script and align the interviewers to seek accuracy and consistency between interviewers (ref), as well as clarity and pertinence of script questions. The analyzes of each city were presented and discussed in a joint workshop with all researchers. The interviews, with an average duration of 50 minutes, were recorded and transcribed in full.

How were the doctors and nurses who worked in the maternity hospitals selected?

We rewrote the paragraph to include information regarding how doctors and nurses were selected: 

“Doctors and nurses who worked in the maternity hospitals were selected based on a nominal list of all professionals who worked in the obstetric hospitalization sectors during the initial period of the pandemic, provided by directors and heads of services. The purposeful sample was defined, seeking to include diverse sociodemographic characteristics, professional experience, and employment relationships”

The analysis section is too brief. 

We have included in the article more details on how the analysis was performed.

I do not understand what this sentence means either – ‘Content analysis was performed in the thematic modality’. 

We used thematic analysis based on two references: Bardin (Content Analysis, 2011, p.199) and Minayo (The challenge of knowledge: qualitative health research, 2014, p. 309) who highlight thematic analysis as one of the ways to perform content analysis. 

I am also not familiar with thematic axes – do you mean themes?

Yes. The expression “thematic axes” has the same meaning as “themes” and we replaced it in the text.

The Discussion section is interesting but I struggled to find a link to the actual findings of the study. I would have liked to see a summary of the findings at the beginning (rather than a rewording of the focus of the study) and then a discussion of how the findings were similar or different to other research. At the moment, the Discussion seems to be quite separate from the study findings. 

We accepted the suggestion and modified the text of the article to address the questions raised by the reviewer 

Reviewer #3: 

The manuscript as a whole should follow standard guidelines for reporting qualitative research as stated in PLOS ONE publication criteria website( https://journals.plos.org/plosone/s/criteria-for-publication) : “Qualitative research studies should be reported in accordance to the Consolidated criteria for reporting qualitative research (COREQ) checklist or Standards for reporting qualitative research (SRQR) checklist. Further reporting guidelines can be found in the Equator Network's Guidelines for reporting qualitative research.” Please re-structure and expand the manuscript to follow the above guidelines. Below are specific recommendations for the manuscript text. 

The standard guidelines for reporting qualitative research used in this article was the Consolidated criteria for reporting qualitative research (COREQ). 

Introduction

While this section provides some background of international healthcare services issues related to covid19, it should be expanded to provide evidence on the context that is specific to Brazil regarding health service disruption, its impact on health workers, and maternal health outcomes during the covid19 pandemic. The evidence provided in the discussion section “Covid-19 and the damage to maternal and child health” such as Tasca et al. 2022, Chisini et al. 2021, and Silva et al., 2021 would be better placed in the introduction so readers outside of Brazil can understand the significance of the study at the start of the manuscript.

We accepted the suggestion and modified the text of the article providing more evidence on the specific context of Brazil in relation to the interruption of health services, its impact on health workers and maternal health outcomes during the Covid-19 pandemic. We have put the authors Tasca et al. 2022, Chisini et al. 2021, and Silva et al., 2021 in the introduction. 

Methods

It seems a survey was only used for the demographic characteristics of participants, but the majority of the data was obtained from in-depth interviews, if so, please clarify this by changing the first sentence of this section from “qualitative exploratory survey” to “qualitative in-depth interviews.”

We accepted the suggestion and modified the text: “Qualitative research with in-depth qualitative interviews.”

It is necessary to use and explain a conceptual or theoretical framework guiding the methods and interpretations of findings (e.g. phenomenology, grounded theory, ethnography, etc.)

We accepted the suggestion and included in the text a statement that the interpretations of the findings were based on comprehensive theory, and we included a reference.

It is necessary to provide more information on the research team and reflexivity, such as but not limited to: which author/s conducted the interviews? What experience or training in research have they received? Did they have any relationship to the study participants prior to the study? 

All interviews were conducted by public health workers with experience in collecting qualitative data and who had no previous relationship with the interviewees. The text of the article was modified to address the questions raised by the reviewer.

“The team of researchers, composed of public health workers, two in São Luís, two in Pelotas, and three in Niterói, all with experience in qualitative data collection, conducted the interviews in person, by telephone, or digital platform, on the days and at the times agreed in advance with the health workers. Workshops were held to prepare and discuss the script, which was the same for the three municipalities, aiming at aligning the interviewers to seek accuracy and consistency among the interviewers. The analyzes of each city were presented and discussed in a joint workshop with all researchers.

Is it possible that the occupation, gender, or any other position of power the interviewer had influenced the answers participants provided or the way the data was collected and analyzed?

The relationship between interviewer and interviewee, in qualitative research, presupposes the existence of asymmetry and social reproduction. Faced with this knowledge, the interviewers sought an empathetic posture that would allow the interviewee to share their experiences without feeling embarrassed or judged. This characteristic of qualitative research was taken into account at the time of the analysis. 

To improve clarity and the flow of ideas, in the last paragraph the sentence in interview duration (lines 160-161) would be best placed on the “Data collection section” and the informed consent sentence (lines 161-162) would be best placed in the “Ethical consideration section:”

All the above considerations were accepted.

Study techniques and tools

⎯ For clarity and avoiding redundancy I suggest removing this sub-heading and move the first paragraph of this section regarding study tools to the “Data Collection” section, and the second paragraph regarding ethics committee approval and confidentiality to the “Ethical consideration” section.

We accepted the suggestions. The subtitle was removed, the duration of the interview was indicated in the data collection session and the second paragraph was inserted in the ethical considerations.

Data collection

⎯ This section needs more detail to assure the reader the research was conducted with scientific rigor.

Please specify the dates data collection started and ended

Data collection started in December 2020 and ended in February 2021, as informed in the first paragraph of Method.

Was the structured questionnaire open answers or multiple choice? 

The questionnaire had open and multiple choice questions and this information was added in the text of the article.

What questions or prompts were used during the interviews?

We have included the main issues of the script in the article.

Was the interview pilot tested?

The interview script was elaborated in a workshop, held among the researchers, to test the clarity and pertinence of this script, making the changes and choices defined by the group of researchers, very similar to the group of interviewees. 

Where interviews recorded and transcribed? 

Yes.

Who did the transcription?

Transcriptions were performed by fellows in the research group and validated by a senior researcher who listened to the recording accompanying the transcribed text.

Where the interviews done in more than one language or just Portuguese? Please state all languages.

All interviews were conducted and transcribed in Portuguese. The fragments used in the article were translated into English.

Where the transcripts returned to participants for any additional comments? 

The transcripts were not returned to respondents. 

Were interview notes taken? Yes. Were they used for the analysis as well?

Notes were taken during the interviews that could facilitate the understanding of the context in which the interviewee’s lines were produced, during the interpretation of the data.

Data analysis

⎯ Please describe how the authors performed the thematic analysis, what steps were taken and by whom? 

The steps for carrying out the thematic analysis were pre-analysis and exploration of the collected material; data processing and interpretation. Therefore, after exhaustive reading of the interviews, we proceeded to categorize the data to extract the relevant themes and then interpret the content, linking the lines with the context of their production. These stages of analysis were initially carried out separately, in each of the databases (São Luís, Niterói, and Pelotas) and coordinated by the responsible researchers in each municipality. In a second moment, web analysis workshops were carried out, to search for confluences and divergences between the results found in the three municipalities. 

- Did more than one person code? If so, how were disagreements resolved? 

Categorization was carried out by more than one researcher and possible divergences were discussed in analysis workshops. In case of doubt, the researchers went back to reading the transcripts and looked for contextual data to support the interpretation. 

- Was any software used or was it done manually? 

No software was used. The analysis matrix was prepared manually by the group of researchers.

- Was there any author reflexivity during the analysis? E.g. How were the researchers assumptions and biases addressed/considered in the analysis?

Considering that there is no neutrality in science, it is always necessary to take into account that researchers’ assumptions and biases can have a significant impact on the results and conclusions of a research. Therefore, we sought to adopt appropriate theoretical-methodological strategies to produce reliable results, namely: be aware of our own beliefs, prejudices and personal experiences that could influence the results; use triangulation to compare and contrast different perspectives and identify possible inconsistencies.

Study participants and sampling

⎯ The phrase “intentionally defined sample” is not common use in Academic English, please correct to” purposeful sampling” if that is what the authors meant.

We accepted the suggestion and modified the text.

⎯ The same for “saturation of the senses”, please correct this to “ data saturation”

We accepted the suggestion and modified the text.

⎯ Please explain why data saturation was used to determine sample size, and how it was determined that saturation had been reach.

Data saturation was used to determine the sample size, as the number of workers at the investigated institutions was large. Sampling closure due to exhaustion (when all participants who meet the inclusion criteria are interviewed), in qualitative research, is indicated only when the total number of participants is small. Therefore, the interviews were interrupted when the answers began to repeat information already obtained, in each institution, according to the saturation criterion, avoiding unnecessary repetitions for the understanding of the object. 

To identify that saturation had been reached, during the fieldwork, workshops were held in each municipality participating in the study (São Luís, Niterói, and Pelotas), from the beginning of data collection, seeking to know and categorize the responses presented. It is known that no line is the same. However, as the collection progressed, the responses began to show common elements. At the beginning of the interviews, new information is evident and then becomes less frequent until it ceases to appear. Interviews were carried out after verifying the repetitions, seeking confirmation of saturation and definition of sample closure. The paragraph has been rewritten to incorporate new information.

⎯ How many participants dropped out before being interviewed? How were they replaced? Where participants interviewed more than once? Was there a follow up interview?

In the case of indirect refusals, the participants were replaced by workers with similar characteristics, based on the nominal list provided by the head of the sector. There was no follow-up interview. Respondents were not interviewed more than once. 

Results

This section needs to be extensively revised to follow best practice when reporting results of a qualitative analysis in academic English. The authors might find the following similar studies useful to re-write this section:

-Leung, C., Olufunlayo, T., Olateju, Z. et al. Perceptions and experiences of maternity care workers during COVID-19 pandemic in Lagos State, Nigeria; a qualitative study. BMC Health Serv Res 22, 606 (2022). https://doi.org/10.1186/s12913-022-08009-y

- Hazfiarini A, Akter S, Homer CSE, Zahroh RI, Bohren MA. ‘We are going into battle without appropriate armour': A qualitative study of Indonesian midwives' experiences in providing maternity care during the COVID-19 pandemic. Women Birth. 2022 Sep;35(5):466-474. doi: 10.1016/j.wombi.2021.10.003. Epub 2021 Oct 11. PMID: 34656517; PMCID: PMC9239738.

Healthcare professionals: sociodemographic characteristics, employment relationship and professional qualifications:

-It is best practice to provide this information in the form of a table and to only summarize in text valuable information (such as the women being majority) or interesting patterns found in this data.

Thanks for the comment. We reviewed the suggested material and adapted the section accordingly.

The religion of the health workers does not seem to be relevant to the research objectives, please remove unless the authors can provide justification of its inclusion in the context of this research study. Removing information about religion.

We agreed with the indication and removed the information about religion from the text.

Thematic axes of statements

⎯ Please state at what point of the research process where the findings translated to English.

The translation into English took place after the final writing and revision of the text in Portuguese.

⎯ Table 1 and table 2: Headers are needed for each column on these tables. It is unclear if each row represents a code, or a higher-level sub-category or sub-theme, please clarify. Additionally, there is repetition of both the content of column one and the column containing quotes, please revise and summarize to avoid redundancy. 

Headings were prepared for each column of the tables and an effort was made to make it clear what the lines represented. In addition, the column content was revised and redundancies were eliminated.

⎯ To improve readability, please add subheadings that represent the subcategories or subthemes under each one of the two major themes 1) changes in hospital organization and dynamics of the pandemic and 2) Illness and suffering of health workers. 

The following subtitles have been added to the themes:

1) Changes in hospital organization and dynamics of the pandemic

Changes in protocols

Between risk and good health practices 

Health care: changes and effects

2) Illness and suffering of health workers

Work overload, lack of workers and their effects

Ambiguities: fear, conflict, cooperation

Risk perception and safety strategies: between work and home

⎯ All statements in the findings need to derive directly from the data collected, therefore, each statement needs to be supported by relevant in-text quotes, context for the findings should be provided in the introduction or methods, and the valuable insights by the authors should be saved for the discussion and backed up by evidence which needs to be referenced. While all this section is very interesting, it is not clear what is commentary by the authors and what was reported by the participants. 

We appreciate the observations and inform you that the reports are related to the perception of the interviewees; we tried to make the context of the discoveries clear in the introduction and leave the interpretation of the data for the discussion

⎯ Line 298 to line 317. Since this is a qualitative study looking at the experiences of health workers, be mindful that the findings cannot state factual increases in service demand. This can be demonstrated by hospital records and other quantitative data, which is beyond the scope of this research. This study can report in the results that participants perceived an increase in demand, and the authors can choose to compare this perception with evidence from other research looking into hospital records and statistics from that period and elaborate on the differences or similarities in the discussion section. 

We thank you for your comments and inform you that the reports are related to the perception of respondents regarding the increased demand in maternity hospitals. The reflections on the decrease in the offer in primary care are also from the health workers participating in the research. The text has been modified to avoid misunderstandings.

Discussion:

- I suggest the authors make use of this guide provided by the journal to revise this section to be more in line to common practice https://plos.org/resource/how-to-write-conclusions/

Thank you for the recommendation.

- The section “social dramas: the pandemic as a total social fact” provides valuable theoretical background which should be summarized and placed in the methods section. 

Our theoretical-methodological proposal consisted of understanding the meanings attributed by subjects in social interaction (Weber, 1993), placing the context as a total social fact (Mauss, 2003), i.e., as a phenomenon that connects several domains such as the social and the individual, on the one hand, and the physical and psychic, on the other. 

- The sections “narratives in dispute” and “covid-19 and the damage to maternal and child health” and “the impact of the pandemic in the medical work” provide relevant context and background to the study, they should be summarized and incorporated to the introduction. 

We accepted the suggestion and modified the text of the article to address the questions raised by the reviewer

- Please revise this section to discuss in depth how the findings of this study specifically compare or relate to the available evidence and social theories provided in the manuscript 

We accepted the suggestion

Reviewer #4:

Abstract

I wonder what precarious work relationships health workers om Niterói experienced compared to São Luís and Pelotas. Would be good to add one or two examples in brackets for this particular sentence (i.e., xxx, xxx). 

We modified the text of the article to address the questions raised by the reviewer, clarifying that, in Niterói, the precariousness of work was related to temporary contracts for workers 

Introduction

In the second paragraph, one would wonder what “in this context” refers to, perhaps paraphrase this to be more clearer? For example, if authors refer to the COVID-19 pandemic, then authors can say 

We accepted the suggestion and modified the text of the article using “During the pandemic”

It would be great if authors can give background information about COVID-19 in Brazil (i.e., burden, trends) to provide context for the readers, and then justify why looking at this in Brazil is important. Adding information on the context of maternal health in Brazil (i.e., use of antenatal care, maternal health indicators – morbidity and mortality, the ratio of health workers to the population) both before and during the pandemic is critical for the readers to be able to interpret the results. I suggest adding this information tool.

We accepted the suggestion and modified the text of the article to address the questions raised by the reviewer 

Providing information on how care changes throughout the pandemic from other settings will be helpful as well to illustrate for readers the kind of information this study was looking for. These two studies have a good example of introduction and will be really great to look at when revising the introduction. 

We read the studies suggested and modified the text of the article 

Methods

Please justify why interview was used compared to other methods (i.e, focus group discussion)

The individual interview was chosen considering that, during the data collection period, there was still a restriction on crowding. In addition, the individual experiences were our greatest interest, reinforced by the fact that they were health workers with different time availability, the individual interview was always the first option. 

Please justify the rationale behind choosing these three municipalities compared to others – what is this sampling based on? Is it because they have the highest COVID-19 rates? Or other reasons?

The suggestion was accepted and we justified the reason for choosing the three municipalities by including a new paragraph in the article and complementing the sixth paragraph informing that the municipalities were chosen because they are located in different regions: the northeast, southeast and south of the country and, therefore, have great diversity of sociodemographic characteristics, different geographic, cultural aspects and in relation to the structure of health services and intersectoral network.

Line 122 what does SUS stand for? Please use full abbreviations for the first use

We modified the text of the introduction to explain SUS as the National Health System in Brazil (SUS - Sistema Único de Saúde)

Are the two hospitals from Pelotas public hospitals? Please state this. And if it is, does that mean all public hospitals? If yes, it’d be good to provide justification for why only public hospitals are included

In Pelotas, there are four hospitals for pregnancy. But only two maternity hospitals were included in this study because only those two hospitals were open during the pandemic to care for patients through SUS.

Other than doctors and nurses, are there any criteria imposed for participation? It’d be great to be clear on the eligibility criteria

For this study, we decided to include only higher-level workers, as they are responsible for defining behaviors. Among these, doctors and nurses were chosen, considering that in the delivery and birth environment, these were the workers who provided assistance. The other higher-level worker categories had been relocated to the care of critically ill patients in the ICU for patients with covid-19. 

Line 146, please add an explanation of how data saturation was determined.

To determine the moment of saturation, from the beginning of data collection, workshops were held in each municipality participating in the study (São Luís, Niterói, and Pelotas), to collect and categorize the responses presented and, therefore, identify the moment when new answers no longer appeared. Interviews were carried out after checking for repetitions, seeking confirmation of saturation and definition of sample closure.

Please add more details on the recruitment. How did the recruitment happen?

As per the suggestion, we have added details regarding recruitment. 

“Doctors and nurses who worked in the maternity hospitals were selected based on a nominal list of all professionals who worked in the obstetric hospitalization sectors during the initial period of the pandemic, provided by directors and heads of services. Characteristics, such as some sociodemographic aspects, professional experience, and employment relationships were considered. The purposeful sampling was defined, seeking to include workers with a diversity of these characteristics. After identifying the workers to be interviewed, contact was made by telephone and/or face-to-face for presenting the research and inviting participation.

Did all doctors and nurses in the three hospitals were approached?

Considering that the number of workers at the investigated institutions was large, the sample closure was not due to exhaustion (when all participants who meet the inclusion criteria are interviewed) but due to saturation. Therefore, the interviews were interrupted when the answers began to repeat information already obtained, in each institution, avoiding unnecessary repetitions for the understanding of the object. The saturation point was determined in workshops in each municipality, during the period of the interviews. Interviews were carried out after the repetitions were checked, seeking confirmation of saturation.” Therefore, a total of 30 health workers were interviewed, 10 in São Luís, 12 in Niterói, and 08 in Pelotas. 

Were advertisement materials circulated?

We did not use advertisement materials. We opted for personal contact with the managers of each of the sectors, asking them to provide a nominal list of workers who worked during the pandemic period, so that we could then choose the interviewees.

How did the authors ensure the sociodemographic of participants are varied? Did this happen before approaching the participant or before? Did all the doctors and nurses in the three hospitals take the demographic survey first and then based on the survey results 30 doctors and nurses approached and recruited?

The elaboration of the nominal list of workers included characteristics, such as professional category, length of service in the hospital, gender, medical or nursing specialty and form of hiring informed by the service manager. After choosing the diversity of these already known characteristics, contact was made to participate in the research. After accepting and signing the TCLE, a structured questionnaire was applied to identify other data related to the sociodemographic profile of the interviewees. In this way, some data were known beforehand and others a posteriori.

When did the survey happen? Is this right before the interview? Or days before the interview itself?

The term survey was inappropriately used when the article was translated into English. We did not conduct a survey. For the identification of sociodemographic characteristics, there was a structured questionnaire completed before the semi-structured interview.

Please add more details on the data collection

The suggestion was accepted, and more details were included in the data collection session in the text of the article.

Please add details on the type of questions/themes asked during the interview and if interview guide/study instruments were piloted. Please attach interview guide/study instruments as an appendix

We added details on the type of questions asked during interviews. It was also clarified that the interview script was tested in a workshop conducted among the researchers and it is included in the annex. 

How many people were involved in the interview? Is it only one person interviewing or there is one interviewer and one observer when the interview happens?

The team of researchers were composed of public health workers, two in São Luís, two in Pelotas, and three in Niterói, all with experience in qualitative data collection, and they conducted the interviews in person, by telephone, or digital platform, on the days and at the times agreed in advance with the health workers. One researcher conducted each interview.

Add information about where the interview was conducted (if it is facility-based or not) and discuss how the different settings might influence the participant's response

In the three municipalities of the study, the interviews were carried out in person, in a reserved room of the service itself, or using the digital platform Google Meet, according to the choice of the interviewee. It is known that different environments can influence the participant’s response, but we always sought to ensure an environment that would allow the confidentiality of responses.

In what language the interview happened? And please add how translations were done if it is in a non-English language

All interviews were conducted, transcribed, and analyzed in Portuguese. Only the excerpts used in the article were translated into English.

Please add more details on the data analysis

The steps for performing the thematic analysis were pre-analysis and exploration of the collected material; data processing and interpretation. Therefore, after exhaustive reading of the interviews, we proceeded to categorize the data to extract the relevant themes and then interpret the content, linking the lines with the context of their production. These stages of analysis were initially carried out separately, in each of the databases (São Luís, Niterói, and Pelotas) coordinated by the researchers responsible for the research in each municipality. In a second moment, web analysis workshops were held, to search for confluences and divergences.

How many researchers were involved in data analysis?

The team of researchers who were involved in data analysis were composed of public health workers, four in São Luís, four in Pelotas, and three in Niterói.

- Was the codebook developed? Please add a coding tree to the manuscript as an appendix.

Yes. A codebook has been developed.

Experience of maternal health workers during the 1st wave of the COVID-19 pandemic

A. Work experience during the pandemic

 1. Changes in work routines (hospital space organization)

 2. Difficulties faced at work (suspension, implementation of services)

 3. Changes in dynamics/pace and intensity of work

B. Perceptions about risk during the pandemic

 1. Adoption of protective measures (means of work - equipment - protocols - training)

 2. relationship between health workers – management and assistance – conflict / cooperation

 3. relationship between health workers – users – family members

 4. Workers’ view on the impact of their work during the pandemic 

C. Illness and suffering during the pandemic 

 1. Work-related illness

 2. Fear, uncertainty, ignorance

 3. work life and private life

D. Perspectives on changes in post-pandemic work

 1. Changes in work organization

 2. Adoption of new technologies and work practices

 3. New perspectives on the role of health workers in society

Were the themes developed inductively or deductively?

The themes were analyzed inductively, with the objective to explore health workers’ perception and identify patterns and relationships between the collected data. There was no prior hypothesis to be tested.

Was any framework used?

Yes. The following framework was used:

Work context during the pandemic (workload, protective measures, changes in work routine, etc.);

Health workers’ perceptions of work during the pandemic;

Strategies used by workers to deal with the challenges of work during the pandemic;

Changes in organization and hospital dynamics in the pandemic

Changes in the work process during the pandemic.

Illness and suffering of health workers;

Implications for professional practice.

Was any data management software (i.e., Nvivo, Atlas) used? 

No

Please add a section on reflexivity, such as information on how interviewer characteristics could influence the participant’s response

The relationship between interviewer and interviewee, in qualitative research, presupposes the existence of asymmetry and social reproduction. Faced with this knowledge, the interviewers sought an empathetic posture that would allow the interviewee to share their experiences without feeling embarrassed or judged. This characteristic of qualitative research was taken into account at the time of the analysis.

Please add details on how many people refuse or drop out of the study

There was no refusal among respondents from Niterói. In Pelotas and São Luís, refusals occurred, and health workers were replaced by others with the same characteristics, based on the nominal list of each sector of the hospital. The following information has been inserted in the article:

“After identifying the workers to be interviewed, contact was made by telephone and/or face-to-face for presenting the research and inviting participation. There were no refusals in Niterói. In São Luís and Pelotas, in the case of indirect refusals, the participants were replaced by workers with similar characteristics, based on the nominal list provided by the head of the sector.”

Please ensure that the reporting of the study adheres to reporting standards for qualitative inquiry, I recommend the use of COREQ reporting standards for qualitative research to ensure compliance with reporting the methods and results of the study: Allison Tong, Peter Sainsbury, Jonathan Craig, Consolidated criteria for reporting qualitative research (COREQ): a 32-item checklist for interviews and focus groups, International Journal for Quality in Health Care, Volume 19, Issue 6, December 2007, Pages 349–357, https://doi.org/10.1093/intqhc/mzm042

We accepted the suggestion and included the following statement: “The article was written based on COREQ recommendations”.

Results

Line 195, please add more details about what precarious work relationships mean and example of this

We modified the text of the article to address the questions raised by the reviewer, clarifying that, in Niterói, the precariousness of work was related to temporary contracts for workers

 Line 197, what is CLT? Please spell out abbreviations on first use 

Ok. We revised all the abbreviations.

I feel that the themes under each thematic axes can be categorized further, this will be really helpful when presenting too so readers will not be lost in words as further categorizations mean that sub-headings under thematic axes can be used, and this can guide readers better. For example, on the thematic axes 1: Changes in hospital organization and dynamics in the pandemic, the themes can be further categorized into services changes, human resource changes, protocol changes, etc. And the results section can be presented using the sub-headings of these categorizations

The following subtitles have been added to the themes:

1) Changes in hospital organization and dynamics of the pandemic

Changes in protocols

Between risk and good health practices 

Health care: changes and effects

2) Illness and suffering of health workers

Work overload, lack of workers and their effects

Ambiguities: fear, conflict, cooperation

Risk perception and safety strategies: between work and home

Please use sub-headings when talking about different themes under the thematic axes to avoid confusion. Another alternative is to bold the specific themes when first use in the specific paragraph, for example: “In Pelotas and São Luís, specialties and elective procedures were suspended. Beds were closed to make entire wards available for the isolation of pregnant and postpartum women. In the three maternity hospitals, healthcare rooms were opened for patients identified as having flu-like symptoms.” I think further categorization of these themes into specific domains will be really helpful for readers as suggested on the previous points

The following subtitles have been added to the themes:

1) Changes in hospital organization and dynamics of the pandemic

Changes in protocols

Between risk and good health practices 

Health care: changes and effects

2) Illness and suffering of health workers

Work overload, lack of workers and their effects

Ambiguities: fear, conflict, cooperation

Risk perception and safety strategies: between work and home

232-233 are too short to stand by themselves as one paragraph, please merge them with another paragraph, same with 219 to 221. 

Suggestion accepted

Please use quotations from participants to illustrate and provide grounds for interpretation. Please see the two studies mentioned in the introduction above for example

Excerpts from the participants’ reports can be found in tables 1 and 2

Discussion

• Before going to the social dramas heading, please add one paragraph summarising the findings of the studies, for example (taken from other study): “Our findings show that midwives in Surabaya and Mataram, Indonesia, strived to deliver maternity care during the COVID-19 pandemic. Despite the difference in the numbers of people with COVID-19 between the study sites, midwives in Surabaya and Mataram shared similar experiences in providing care, except for an increase in workload, which was mostly faced by midwives in Surabaya.” (Hazfiarini, 2021)

Suggestion accepted

• The notion of social fact by Marcel Mauss is very interesting and it will enrich the discussion if authors can expand/connect the social fact define in the discussion with the study findings, and what lesson can be taken to prepare better for future pandemics 

Suggestion accepted

• Again, for narratives in dispute, what lessons can be taken from the study results? What can we do better for the health workers? 

Suggestion accepted

• Please add the strengths and limitations of the studies as “Strengths and limitations” section, and I think the strengths aspects can be expanded more Please add how the authors have addressed the identified limitations.

Suggestion accepted

• Another aspect of the study is to discuss the “transferability” of the results to other settings in Brazil or beyond

Suggestion accepted

Besides, we removed the funding information from the Acknowledgments section.

---

## [Decision Letter · Decision Letter 1]

23 May 2023

PONE-D-22-34357R1Maternal health during the COVID-19 pandemic: experiences of health workers in three Brazilian municipalitiesPLOS ONE

Dear Dr. Carvalho,

Thank you for submitting your manuscript to PLOS ONE. After careful consideration, we feel that it has merit but does not fully meet PLOS ONE’s publication criteria as it currently stands. Therefore, we invite you to submit a revised version of the manuscript that addresses the points raised during the review process.

We look forward to receiving your revised manuscript.

Kind regards,

Kyaw Lwin Show, MPH

Academic Editor

PLOS ONE

Journal Requirements:

Reviewers' comments:

Reviewer's Responses to Questions

**Comments to the Author**

1. If the authors have adequately addressed your comments raised in a previous round of review and you feel that this manuscript is now acceptable for publication, you may indicate that here to bypass the “Comments to the Author” section, enter your conflict of interest statement in the “Confidential to Editor” section, and submit your "Accept" recommendation.

Reviewer #3: (No Response)

Reviewer #4: All comments have been addressed

2. Is the manuscript technically sound, and do the data support the conclusions?

Reviewer #3: Partly

Reviewer #4: Yes

3. Has the statistical analysis been performed appropriately and rigorously? 

Reviewer #3: N/A

Reviewer #4: N/A

4. Have the authors made all data underlying the findings in their manuscript fully available?

Reviewer #3: Yes

Reviewer #4: Yes

5. Is the manuscript presented in an intelligible fashion and written in standard English?

Reviewer #3: Yes

Reviewer #4: Yes

6. Review Comments to the Author

Reviewer #3: Thank you for your responses and modification of the manuscript.

In the findings section: Please integrate the quotes in the tables to the text on each theme as shown in the examples provided by reviewers in the previous round of comments. Integrating relevant quotes after each statement in the findings makes it possible for the reader to understand how those statements are supported by the primary data and how the conclusions related to the data were drawn.

Reviewer #4: Thank you for addressing my comments. This is a well conducted and written study, and I have no further comments.

7. PLOS authors have the option to publish the peer review history of their article (what does this mean?). If published, this will include your full peer review and any attached files.

Reviewer #3: No

Reviewer #4: No

---

## [Author Response · Author response to Decision Letter 1]

31 Jul 2023

Dear Academic Editor and Reviewers,

Thank you for the evaluation of the above-mentioned manuscript. 

After careful reading, we considered suggestion made and responded the comment/suggestion in this document. We are resubmitting the manuscript to Plos One. 

In addition to the ‘Rebuttal letter’, we have also included a ‘Revised Manuscript with Changes Highlighted’ and an unmarked version of the paper without tracked changes, according to the instructions received.

Thank you very much for the opportunity to review our work.

 Kind regards,

 The Authors

6. Review Comments to the Author

Reviewer #3: Thank you for your responses and modification of the manuscript.

In the findings section: Please integrate the quotes in the tables to the text on each theme as shown in the examples provided by reviewers in the previous round of comments. Integrating relevant quotes after each statement in the findings makes it possible for the reader to understand how those statements are supported by the primary data and how the conclusions related to the data were drawn.

We accepted the suggestion and modified the text to address thr questions raised by the reviewer. We appreciate the comments and inform you that the interviewees' speeches were inserted in the text for greater/better understanding of the context and our analyses.

We informed you that the phrase has been moved: The Niterói maternity hospital offered testing for COVID-19 (rapid test and PCR). (line 423-424)

In this sentence, we clarify that health professionals mentioned a behavior of pacientes and family members: In Niterói, difficulty in adherence to preventive measures by patients and their families was also mentioned. (line 404-405)

Correction: In the four maternity (line 557)

---

## [Decision Letter · Decision Letter 2]

2 Aug 2023

Maternal health during the COVID-19 pandemic: experiences of health workers in three Brazilian municipalities

PONE-D-22-34357R2

Dear Dr. Carvalho,

We’re pleased to inform you that your manuscript has been judged scientifically suitable for publication and will be formally accepted for publication once it meets all outstanding technical requirements.

Kind regards,

Kyaw Lwin Show, MPH

Academic Editor

PLOS ONE

Additional Editor Comments (optional):

Reviewers' comments:

Reviewer's Responses to Questions

**Comments to the Author**

1. If the authors have adequately addressed your comments raised in a previous round of review and you feel that this manuscript is now acceptable for publication, you may indicate that here to bypass the “Comments to the Author” section, enter your conflict of interest statement in the “Confidential to Editor” section, and submit your "Accept" recommendation.

Reviewer #3: All comments have been addressed

2. Is the manuscript technically sound, and do the data support the conclusions?

Reviewer #3: Yes

3. Has the statistical analysis been performed appropriately and rigorously? 

Reviewer #3: N/A

4. Have the authors made all data underlying the findings in their manuscript fully available?

Reviewer #3: No

5. Is the manuscript presented in an intelligible fashion and written in standard English?

Reviewer #3: Yes

6. Review Comments to the Author

Reviewer #3: (No Response)

7. PLOS authors have the option to publish the peer review history of their article (what does this mean?). If published, this will include your full peer review and any attached files.

Reviewer #3: No

---

## [Editor Report · Acceptance letter]

21 Aug 2023

PONE-D-22-34357R2 

Maternal health during the COVID-19 pandemic: experiences of health workers in three Brazilian municipalities 

Dear Dr. Carvalho:

I'm pleased to inform you that your manuscript has been deemed suitable for publication in PLOS ONE. Congratulations! Your manuscript is now with our production department. 

Kind regards, 

on behalf of

Dr. Kyaw Lwin Show 

Academic Editor

PLOS ONE